# Query-Efficient Correlation Clustering with Noisy Oracle

**Yuko Kuroki**
CENTAI Institute
Turin, Italy
yuko.kuroki@centai.eu

**Atsushi Miyauchi**
CENTAI Institute
Turin, Italy
atsushi.miyauchi@centai.eu

**Francesco Bonchi**
CENTAI Institute, Turin, Italy
Eurecat, Barcelona, Spain
bonchi@centai.eu

**Wei Chen**
Microsoft Research
Beijing, China
weic@microsoft.com

## Abstract

We study a general clustering setting in which we have $n$ elements to be clustered, and we aim to perform as few queries as possible to an oracle that returns a noisy sample of the weighted similarity between two elements. Our setting encompasses many application domains in which the similarity function is costly to compute and inherently noisy. We introduce two novel formulations of online learning problems rooted in the paradigm of Pure Exploration in Combinatorial Multi-Armed Bandits (PE-CMAB): fixed confidence and fixed budget settings. For both settings, we design algorithms that combine a sampling strategy with a classic approximation algorithm for correlation clustering and study their theoretical guarantees. Our results are the first examples of polynomial-time algorithms that work for the case of PE-CMAB in which the underlying offline optimization problem is NP-hard.

## 1 Introduction

Given a set $V = [n]$ of $n$ objects and a pairwise similarity measure $\mathrm{s} : \binom{V}{2} \to [0, 1]$ (where $\binom{V}{2}$ is the set of unordered pairs of elements of $V$, and the value closer to $1$ means higher similarity), the goal of *Correlation Clustering* [7] is to cluster the objects so that, to the best possible extent, similar objects are put in the same cluster and dissimilar objects are put in different clusters. Assuming that cluster identifiers are represented by natural numbers, a clustering $\mathcal{C}$ can be represented as a function $\ell : V \to \mathbb{N}$, where each cluster is a maximal set of objects sharing the same label. The objective is to minimize the following cost:

$$\mathrm{cost}_{\mathrm{s}}(\ell) = \sum_{\substack{(x,y) \in \binom{V}{2}, \\ \ell(x) = \ell(y)}} (1 - \mathrm{s}(x, y)) + \sum_{\substack{(x,y) \in \binom{V}{2}, \\ \ell(x) \neq \ell(y)}} \mathrm{s}(x, y). \tag{1}$$

The intuition underlying the above problem definition is that if two objects $x$ and $y$ are dissimilar, expressed by a small value of $\mathrm{s}(x, y)$, yet they are assigned to the same cluster, we should incur a high cost. Conversely, if $\mathrm{s}(x, y)$ is high, indicating that $x$ and $y$ are very similar, but they are assigned to different clusters, we should also incur a high cost.

Two key features make correlation clustering quite suitable in real-world applications. Firstly, it does not require the number of clusters as part of the input; instead, it automatically finds the optimal number, performing model selection. Secondly, it only requires the pairwise information without

38th Conference on Neural Information Processing Systems (NeurIPS 2024).

assuming any specific structure of the data. This reasonably eliminates the need for domain knowledge about complex data. Correlation clustering has been applied to a wide range of problems across various domains, including duplicate detection and similarity joins [34, 46], spam detection [12, 72], co-reference resolution [66], biology [9, 14], image segmentation [54], social network analysis [13], and clustering aggregation [42].

Correlation clustering is NP-hard even in the simplest formulations [7, 75], and minimizing the cost function in (1) is APX-hard [20]; thus, we cannot expect a polynomial-time approximation scheme. Nevertheless, there are a number of constant-factor approximation algorithms for various settings [1, 3, 7, 20, 22, 30–32]. For the formulation of (1), Ailon et al. [3] presented KwikCluster, a simple 5-approximation algorithm. The algorithm randomly picks a *pivot* $v \in V$ and constructs a cluster by taking all the vertices *similar* to $v$; then, the algorithm removes the cluster and repeats the process until $V$ is fully clustered. The simplicity and theoretical guarantees of KwikCluster have produced a lot of variations in different scenarios [13, 29, 61, 69, 76, 79].

In practice, preparing the similarity function involves *costly measurements*. Given $n$ items to be clustered, $\Theta(n^2)$ similarity computations are needed to prepare the input to correlation clustering algorithms. Moreover, computing the similarity $s(x, y)$ might have additional expenses (e.g., human effort or financial resources) besides the mere computational cost. To mitigate these issues, some query-efficient methods have been proposed based on the active learning framework [11, 15, 40]. In this framework, the similarity function is initially unknown but an oracle that returns the true similarity in $\{0, 1\}$ for a pair of objects is sequentially queried. In particular, these studies provided a randomized algorithm that, given a budget $T$ of queries, attains a solution whose expected cost is at most $3 \cdot \text{OPT} + \mathcal{O}(\frac{n^3}{T})$, where OPT is the optimal value of the problem.

However, the above methods for query-efficient correlation clustering have significant limitations. Indeed, all the aforementioned works [11, 15, 40] only consider the *binary similarity* of $\{0, 1\}$, while the similarity between two objects are often non-binary in real-world scenarios. For example, in biological sciences, protein-protein interaction networks are commonly analyzed, where the strength of the interactions among proteins is represented as a real-valued similarity [68]. As another example, in entity resolution, i.e., a task central to data integration [80], real-valued similarity is used to indicate the likelihood of matches of objects instead of binary decisions. Therefore, allowing the similarity to be real-valued in the interval $[0, 1]$ would be more practical and flexible. Furthermore, the above works assume the access to the *strong oracle* that returns the true value of $s(x, y)$ ($= 0$ or $1$), while evaluating $s(x, y)$ might be inherently *noisy*, due to error-prone experiments, noisy measurements, or biased judgments. In the above first example the strength of the interactions among proteins is often measured based on biological experiments involving unavoidable noise, while in the second example the likelihood of matches of objects is usually obtained based on biased human judgements.

In this paper, we focus on the challenging scenario where (i) the underlying similarity measure can take any real value in $[0, 1]$ rather than being binary, and (ii) we can only query a noisy oracle that provides inaccurate evaluations of the weighted similarity $s(x, y)$. The goal of this paper is *to devise clustering algorithms that perform as few queries on $s(x, y)$ as possible to an oracle that returns noisy answers to $s(x, y)$*. In pursuit of this goal, we introduce two novel formulations based on multi-armed bandits problems, both of which achieve a reasonable trade-off between the number of queries to the oracle and the quality of solutions.

While our problem formulations are novel, recent prior work has explored related issues. Silwal et al. [76] proposed a practical model using the strong oracle along with a cheaper but inaccurate oracle. Their algorithm achieves a cost of $3 \cdot \text{OPT} + \epsilon n^2$ using $n + \mathcal{O}(\frac{\gamma}{\epsilon})$ queries to the strong oracle, where $\gamma > 0$ is the error level of noisy oracle and $\epsilon > 0$ is the additive error. However, they still focus on the binary similarity and there is no guarantee on the query upper bound for the noisy oracle. Unlike theirs, our models are designed to handle the weighted similarity and do not rely on any strong oracle. Aronsson and Chehreghani [4, 5] studied a non-persistent noise model where the oracle returns the true value of $s(x, y)$ with probability $1 - \gamma$ and a noisy value otherwise. Their algorithm handles a general weighted similarity but provides neither query complexity nor approximation guarantee.

## 1.1 Our contributions

In this paper, we study the problem of *query-efficient correlation clustering with noisy oracles*, where the similarity function $s : \binom{V}{2} \to [0, 1]$ is *initially unknown*, and only *noisy feedback* instead of

the true similarity $s(x, y)$ is observed when querying a pair of objects $(x, y)$. In this scenario, it is desired to achieve a reasonable trade-off between the number of queries to the oracle and the cost of clustering. To this end, we introduce two formulations of online learning problems rooted in the paradigm of *Pure Exploration of Combinatorial Multi-Armed Bandits* (PE-CMAB). In the *fixed confidence setting* (Problem 1), given a confidence level $\delta \in (0, 1)$, the learner aims to find a well-approximate solution with probability at least $1 - \delta$ while minimizing the number of queries required to determine the output. Conversely, in the *fixed budget setting* (Problem 2), given a querying budget $T$, the learner aims to maximize the probability that the output is a well-approximate solution. Our contributions can be summarized as follows:

- For Problem 1, we design KC-FC (Algorithm 1), which effectively combines *threshold bandits* with KwikCluster. We prove that given confidence level $\delta \in (0, 1)$, KC-FC finds a solution whose expected cost is at most $5 \cdot \mathrm{OPT} + \epsilon$ with probability at least $1 - \delta$, where OPT is the optimal value of the problem, and provide the upper bound of the number of queries (Theorem 1).
- We design KC-FB (Algorithm 3) for Problem 2, which adaptively determines the number of queries for each pair of objects based on KwikCluster. We prove that the error probability of the expected cost being worse than $5 \cdot \mathrm{OPT} + \epsilon$ decreases exponentially with budget $T$ (Theorem 2).
- We empirically validate our theoretical findings by demonstrating that KC-FC and KC-FB outperform baseline methods in terms of the sample complexity and cost of clustering, respectively.

It is worth noting that our approximation guarantees in Theorems 1 and 2 match the approximation ratio 5 of KwikCluster [3], where $s : \binom{V}{2} \to [0, 1]$ is known in advance, up to the additive error $\epsilon > 0$. These results are not achievable using existing PE-CMAB algorithms due to the NP-hardness of correlation clustering. In the standard PE-CMAB, a learner aims to identify the best action that maximizes the linear reward from the combinatorial decision set $\mathcal{D} \subseteq 2^{[m]}$ with $m$-base arms. Existing algorithms for PE-CMAB (e.g., [23, 25, 35, 52, 82]) rely on the assumption that the offline problem is polynomial-time solvable. Redesigning them to obtain a well-approximate solution while running efficiently is quite challenging, as the exact optimization of the offline problem is crucial to achieving statistical validity and a correctness guarantee for the output. Ours are the first polynomial-time algorithms that work for the case of PE-CMAB where the underlying offline optimization is NP-hard, filling a critical gap in existing PE-CMAB algorithms, which is of independent interest.

## 1.2 Related work

**Correlation clustering with noisy input.** The bulk of the literature on noisy correlation clustering (see Section 4.6 of Bonchi et al. [10]) considers the binary similarity and assumes that there is the ground-truth clustering but some of the $s(x, y)$ are wrong: they are 0 instead of 1, or vice versa. The seminal work by Bansal et al. [7] and Joachims and Hopcroft [49] provided the bounds on the error with which correlation clustering recovers the ground truth under a simple probabilistic model over graphs. Mathieu and Schudy [64] studied the model starting from an arbitrary partition of the $n$ elements into clusters, where $s(x, y)$ is perturbed independently with probability $p$, and a more general model with the adversary. They proposed an algorithm that achieves some approximation ratio and manages to approximately recover the ground truth. Chen et al. [28] extended the framework to sparse Erdős–Rényi random graphs and obtained an algorithm that conditionally recovers the ground truth. Finally, Makarychev et al. [62] overcame some limitations of Mathieu and Schudy [64] and Chen et al. [28]; they assumed very little about the observations and gave two approximation algorithms. Unlike the above models, ours is based on online learning with an unknown distribution with mean of $s(x, y)$, which is in general not binary, and does not assume any ground-truth clustering.

**Combinatorial multi-armed bandits.** *Multi-Armed Bandit* (MAB) is a classical decision-making model [57, 59, 73]: There are $m$ possible actions (called *arms*), whose expected reward $\mu_i$ for each $i \in [m]$ is unknown. At each round, a learner chooses an arm to pull and observes a stochastic reward sampled from an unknown probability distribution. The most popular objective is to minimize the cumulative regret [16, 18]. Another popular objective is to identify the arm with the maximum expected reward. This problem, called the *Best Arm Identification* (BAI) or *Pure Exploration* (PE) in MAB, has also received much attention [6, 8, 17, 21, 24, 37, 38, 41, 48, 53]. The model of *Combinatorial Multi-Armed Bandits* (CMAB) is a generalization of MAB [19, 26], where an interested subset of arms forms a certain combinatorial structure such as a spanning tree, matching, or path. Since its introduction by Chen et al. [25], the study of PE-CMAB has been actively pursued

in various settings [23, 35, 36, 47, 50, 52, 55, 56, 67, 78, 82]. Notably, Gullo et al. [44] addressed regret minimization for correlation clustering by adapting UCB-type algorithms. However, regret minimization in CMAB is quite different from pure exploration framework when working with approximation oracles (i.e., offline approximation algorithms) for solving NP-hard problems. For regret minimization, we can incorporate approximation oracles with the UCB framework, consistent with the optimization under uncertainty principle (e.g., [26, 27, 81]). However, in pure exploration, the lack of uniqueness of $\alpha$-approximate solutions makes it difficult to determine the stopping condition in the FC setting. In the FB setting, the Combinatorial Successive Accept Reject algorithm proposed by Chen et al. [25] iteratively solves the so-called Constrained Oracle problem, which is often NP-hard, as later addressed in Du et al. [35]. We anticipate a similar NP-hard problem in correlation clustering, requiring a different approach.

**Other clustering settings.** Ailon et al. [2] and Saha and Subramanian [74] studied correlation clustering with *same-cluster queries*, where all similarities of $\binom{V}{2}$ are known in advance and their query is further allowed to access the optimal clustering. Our setting differs significantly as we are interested in the case where similarities are unknown and only noisy similarity values are received rather than same-cluster queries. Finally, it is worth mentioning that Xia and Huang [83] and Gupta et al. [45] proposed a MAB approach for clustering reconstruction with noisy same-cluster queries [58, 65, 70, 71, 77]. However, this clustering reconstruction problem does not directly offer any algorithmic result for correlation clustering. The detailed comparison is deferred to Appendix A.

## 2 Problem statements

Here we formally define our formulations of PE-CMAB for correlation clustering. Our problem instances are characterized by $(V, \mathrm{s})$, where $V = [n]$ is the set of elements to be clustered and $\mathrm{s} : \binom{V}{2} \to [0, 1]$ is the pairwise similarity function, which is *unknown* to the learner. Define the set of unordered pairs as $E = \binom{V}{2}$ with $m := |E|$.

At each round $t = 1, 2, \ldots$, a learner will pull (i.e., query) one arm (i.e., pair of elements in $V$) from action space $E = \binom{V}{2}$ based on past observations. After pulling $e \in E$, the learner can observe the random feedback $X_t(e)$, which is independently sampled from an *unknown* distribution such as Bernoulli or $R$-sub-Gaussian with unknown mean $\mathrm{s}(e) \in [0, 1]$.[1] After some exploration rounds, the learner must identify a well-approximate solution. Let $\mathrm{OPT}(\mathrm{s})$ be the optimal value of the offline problem minimizing the cost function (1) and let $\mathcal{C}_{\mathrm{out}}$ be the output by an algorithm. For $\alpha \geq 1$ and $\epsilon > 0$, we say $\mathcal{C}_{\mathrm{out}}$ to be an $(\alpha, \epsilon)$-approximate solution if $\mathrm{cost}_{\mathrm{s}}(\mathcal{C}_{\mathrm{out}}) \leq \alpha \cdot \mathrm{OPT}(\mathrm{s}) + \epsilon$. We study the following two formulations: Fixed Confidence (FC) and Fixed Budget (FB) settings.

**Problem 1** (Fixed confidence setting)**.** *Let $\alpha \geq 1$. Given a confidence level $\delta \in (0, 1)$ and additive error $\epsilon > 0$, the learner aims to guarantee that the output $\mathcal{C}_{\mathrm{out}}$ is an $(\alpha, \epsilon)$-approximate solution with probability at least $1 - \delta$. The evaluation metric of an algorithm is the sample complexity, i.e., the number of queries to the oracle the learner uses.*

**Problem 2** (Fixed budget setting)**.** *Let $\alpha \geq 1$. Given a querying budget $T$ and additive error $\epsilon > 0$, the learner aims to maximize the probability that the output $\mathcal{C}_{\mathrm{out}}$ is an $(\alpha, \epsilon)$-approximate solution.*

Note that the case of $\alpha = 1$ corresponds to the standard PE-CMAB formulations. However, as the offline problem minimizing the cost function (1) is APX-hard [20], we cannot expect any polynomial-time algorithm that can handle $\alpha = 1$ in the above formulations.

## 3 Fixed confidence setting

In this section, we design KC-FC (Algorithm 1) for Problem 1, built on a novel combination of KwikCluster (detailed in Algorithm 4 in Appendix B) and techniques of threshold bandits. The key idea of the proposed method is to first identify pairs with seemingly high similarity, which are then passed to KwikCluster to produce a high-quality clustering.

---

[1] We use Bernoulli distribution for the sake of simplicity, i.e., $X_t(e) \sim \mathrm{Bern}(\mathrm{s}(e))$, where $\mathrm{s}(e)$ is the unknown mean. We can consider $R$-sub-Gaussian distribution and our results carry on, by simply adjusting the statement of the Hoeffding inequality accordingly.

---

**Algorithm 1** KwikCluster with Fixed Confidence (KC-FC)

---

**Input** : Confidence level $\delta$, set $V$ of $n$ objects, and error $\epsilon$

1 $E_1 \leftarrow E$, $V_1 \leftarrow V$, $r \leftarrow 1$, and $\mathcal{C}_{\text{out}} \leftarrow \emptyset$;

2 Compute $\widehat{G}_{\epsilon'}$ by TB-HS (Algorithm 2) with $\epsilon' = \frac{\epsilon}{12m}$;

3 Define $\widehat{\Gamma}(v) := \{u \in V : \{u, v\} \in \widehat{G}_{\epsilon'}\}$;

4 **while** $|V_r| > 0$ **do**

5     Pick a pivot $p_r \in V_r$ uniformly at random;

6     $\mathcal{C}_{\text{out}} \leftarrow \mathcal{C}_{\text{out}} \cup \{C_r\}$, where $C_r := (\{p_r\} \cup \widehat{\Gamma}(p_r)) \cap V_r$;

7     $V_{r+1} \leftarrow V_r \setminus C_r$ and $r \leftarrow r + 1$;

8 **return** $\mathcal{C}_{\text{out}}$

---

For the first phase, we leverage one of the variants of MAB, called the *threshold bandits* [51, 60, 63], which is defined as follows: Given a confidence level $\delta$ and $m$-arms, the learner must return the set of *good* arms, i.e., arms whose expected rewards are greater than a given threshold $\theta > 0$, as soon as possible, and stops when the learner believes that there is no remaining good arm, w.p. at least $1 - \delta$. TB-HS (detailed in Algorithm 2) is our key procedure, which is designed for identifying seemingly high similarity pairs. Note that, if we naively use the existing algorithm by Kano et al. [51] for threshold bandits where the set of arms is $E = \binom{V}{2}$ and the threshold is $\theta = 0.5$, the algorithm is not even guaranteed to terminate; the resulting sample complexity becomes infinitely large if $\text{s}(e) = \text{s}(e')$ for different $e, e' \in E$ or if there exists $e \in E$ with $\text{s}(e) = 0.5$, which may frequently happen in practice. Our strategy to avoid such an unbounded sample complexity is to allow TB-HS to misidentify pairs of elements with similarity close to 0.5, taking advantage of the fact that the output accuracy can be guaranteed despite such misidentification.

**Algorithm details.** Let $\widehat{\text{s}}_t(e)$ be the empirical mean of the similarity for each pair $e \in E$ kept at round $t$. Let $N_t(e)$ be the number of queries of $e \in E$ that has been pulled by the end of round $t$. TB-HS maintains the confidence bound defined as $\text{rad}_t(e) := \sqrt{\frac{\log(4mN_t(e)^2/\delta)}{2N_t(e)}}$ for each $e \in E$. The arm selection at round $t$ is based on the Lower-Confidence-Bound (LCB) score, i.e., $\underline{\text{s}}_t(e) := \widehat{\text{s}}_t(e) - \text{rad}_t(e)$ and the Upper-Confidence-Bound (UCB) score, i.e., $\bar{\text{s}}_t(e) := \widehat{\text{s}}_t(e) + \text{rad}_t(e)$. We pull the arm $\hat{e}_t^g$ with the highest LCB (line 5) and the arm $\hat{e}_t^b$ with the lowest UCB (line 6). Then $\hat{e}_t^g$ will be added to $\widehat{G}_\epsilon$ if its LCB is no less than $0.5 - \epsilon$, and $\hat{e}_t^b$ will be added to $\widehat{B}_\epsilon$ if its UCB is no greater than $0.5 + \epsilon$. TB-HS continues this procedure until every $e \in E$ is added to either $\widehat{G}_\epsilon$ or $\widehat{B}_\epsilon$. Our main algorithm KC-FC invokes TB-HS to compute $\widehat{G}_{\epsilon'}$ with parameter $\epsilon' = \frac{\epsilon}{12m}$. Then it carries out KwikCluster using the predicted similarity by $\widehat{G}_{\epsilon'}$ as follows. Until an unclustered element exists, it picks one pivot element $p_r$ uniformly at random, builds a cluster $C_r$ around it by adding those among the unclustered elements that seemingly have high similarity with a pivot $p_r$ (based on $\widehat{G}_{\epsilon'}$), and removes all the elements in $C_r$ from the list of unclustered elements.

**Analysis.** For a given $\epsilon \in (0, 0.5)$, we define the following sets, which appear only in the theoretical analysis and are unknown to the learner: $E_{[0.5\pm\epsilon]} := \{e \in E : |0.5 - \text{s}(e)| \leq \epsilon\}$, $E_{(0.5+\epsilon,1]} := \{e \in E : \text{s}(e) > 0.5 + \epsilon\}$, and $E_{[0,0.5-\epsilon)} := \{e \in E : \text{s}(e) < 0.5 - \epsilon\}$. For $\epsilon \in (0, 0.5)$, we introduce the definition of the gaps that characterize our sample complexity:

$$\tilde{\Delta}_{e,\epsilon} := \left(\Delta_e + \min\left\{\epsilon - \Delta_{\min}, \frac{\epsilon}{2}\right\}\right) \text{ for } e \in [m], \tag{2}$$

where $\Delta_e := |\text{s}(e) - 0.5|$ for $e \in [m]$ and $\Delta_{\min} := \min_{e \in [m]} \Delta_e$.

Now we present our theorem, guaranteeing that KC-FC finds a $(5, \epsilon)$-approximate solution with high probability and provides an upper bound of the number of queries, i.e., the sample complexity.

**Theorem 1.** *Given a confidence level $\delta \in (0, 1)$ and additive error $\epsilon > 0$, KC-FC (Algorithm 1) guarantees that*

$$\Pr[\text{cost}_\text{s}(\mathcal{C}_{\text{out}}) \leq 5 \cdot \text{OPT}(\text{s}) + \epsilon] \geq 1 - \delta,$$

**Algorithm 2** Threshold Bandits for indentifying High Similarity pairs with $\epsilon \in (0, 0.5)$ (TB-HS).

**Input** : Set $E$ of $m$-arms and confidence level $\delta$

1   $\widehat{G}_\epsilon \leftarrow \emptyset$ and $\widehat{B}_\epsilon \leftarrow \emptyset$;

2   Pull each $e \in E$ once to initialize empirical mean $\widehat{s}_m(e)$, $t \leftarrow m$, and $E_t \leftarrow E$;

3   Compute $\mathrm{rad}_t(e) := \sqrt{\frac{\log(4m N_t(e)^2/\delta)}{2 N_t(e)}}$ for $e \in E_t$;

4   **while** $|E_t| > 0$ **do**

5      Pull $\hat{e}_t^g := \mathrm{argmax}_{e \in E_t}(\widehat{s}_t(e) - \mathrm{rad}_t(e))$ once;

6      Pull $\hat{e}_t^b := \mathrm{argmin}_{e \in E_t}(\widehat{s}_t(e) + \mathrm{rad}_t(e))$ once;

7      Update $\widehat{s}_t$ and $\mathrm{rad}_t$ for $\hat{e}_t^g$ and $\hat{e}_t^b$;

8      **if** $\underline{s}_t(\hat{e}_t^g) := \widehat{s}_t(\hat{e}_t^g) - \mathrm{rad}_t(\hat{e}_t^g) \geq 0.5 - \epsilon$ **then**

9         Add $\hat{e}_t^g$ to good arms, i.e., $\widehat{G}_\epsilon \leftarrow \widehat{G}_\epsilon \cup \{\hat{e}_t^g\}$, and delete $\hat{e}_t^g$ from $E_t$;

10     **if** $\bar{s}_t(\hat{e}_t^b) := \widehat{s}_t(\hat{e}_t^b) + \mathrm{rad}_t(\hat{e}_t^b) \leq 0.5 + \epsilon$ **then**

11        Add $\hat{e}_t^b$ to bad arms, i.e., $\widehat{B}_\epsilon \leftarrow \widehat{B}_\epsilon \cup \{\hat{e}_t^b\}$, and delete $\hat{e}_t^b$ from $E_t$;

12     $E_{t+2} \leftarrow E_t$;

13     $t \leftarrow t + 2$;

14   **return** $\widehat{G}_\epsilon$

---

*and letting $\epsilon' = \frac{\epsilon}{12m}$, the sample complexity $T$ is*

$$\mathcal{O}\left( \sum_{e \in E} \frac{1}{\tilde{\Delta}_{e,\epsilon'}^2} \log\left( \frac{n}{\tilde{\Delta}_{e,\epsilon'}^2 \delta} \log\left( \frac{n}{\tilde{\Delta}_{e,\epsilon'}^2 \delta} \right) \right) + \frac{n^2}{\max\left\{ \Delta_{\min}, \frac{\epsilon'}{2} \right\}^2} \right).$$

*Furthermore, KC-FC runs in time polynomial in $n$.*

*Proof Sketch.* For the outputs $\widehat{G}_\epsilon$ and $\widehat{B}_\epsilon$ of TB-HS (Algorithm 2) with parameters $\epsilon \in (0, 0.5)$ and $\delta \in (0, 1)$, by using the Hoeffding inequality and the procedure of TB-HS (lines 8 and 10), it is easy to see that $E_{(0.5+\epsilon, 1]} \subseteq \widehat{G}_\epsilon$ and $E_{[0, 0.5-\epsilon)} \subseteq \widehat{B}_\epsilon$ w.p. at least $1 - \delta$. Consider the similarity function $\widetilde{s} : E \to [0, 1]$ such that for each $e \in E$, $\widetilde{s}(e) = s(e)$ if $e \in E_{[0, 0.5-\epsilon)} \cup E_{(0.5+\epsilon, 1]}$, and $\widetilde{s}_e$ otherwise, where $\widetilde{s}_e$ is an arbitrary value that satisfies $|s(e) - \widetilde{s}_e| < 2\epsilon$. Noticing that KC-FC corresponds to KwikCluster associated with a certain choice of $\widetilde{s}$ (i.e., $\widetilde{s}_e$ for $e \in E_{[0.5 \pm \epsilon]}$), we can show that $\mathbb{E}[\mathrm{cost}_s(\mathcal{C}_{\mathrm{out}})] \leq 5 \cdot \mathrm{OPT}(s) + 12\epsilon |E_{[0.5 \pm \epsilon]}|$ for the output $\mathcal{C}_{\mathrm{out}}$, providing the approximation guarantee. The rest of the proof requires the analysis of the upper bound of the number of queries that TB-HS used to stop. This can be done based on a prior analysis of threshold bandits [51], while carefully handling $\epsilon > 0$. The complete proof for analysis is given in Appendix C.

For the time complexity, each iteration of sub-routine TB-HS takes $\mathcal{O}(m)$ steps in a naive implementation or amortized $\mathcal{O}(\log T)$ steps if we manage arms using two heaps corresponding to LCB/UCB values, and the other procedure in KC-FC runs in time polynomial in $n$. □

**Comparison with existing PE-CMAB methods in the FC setting.** Existing methods for PE-CMAB (e.g., [23, 25, 35, 82]) are limited by their reliance on the polynomial-time solvability of the underlying offline problem. If we use an efficient approximation algorithm in those existing methods, their stopping conditions no longer have a guarantee of the quality of the output. Specifically, such existing methods use the LUCB-type strategy, and its stopping condition requires the exact computation of the empirical best solution and the second empirical best solution to check if the current estimation is enough or not. When we only have an approximate oracle (i.e., approximation algorithm), such existing stopping conditions are no longer valid, and the algorithm is not guaranteed to stop. In contrast, KC-FC runs in time polynomial in $n$ while ensuring sample complexity and approximation guarantee. We also note that $\Delta_e$, the distance between $s(e)$ and $0.5$, interestingly characterizes our sample complexity, as we show that the learning task boils down to identifying $E_{(0.5+\epsilon', 1]}$ and $E_{[0, 0.5-\epsilon')}$ thanks to the behavior of KC-FC – they leverage the property that by accurately estimating the mean of the base arms (i.e., pairs of elements), we can maintain the approximation guarantee of KwikCluster in the offline setting with small additive error.

**Statistical efficiency.** In the noise-free setting, $\binom{n}{2}$ queries are sufficient, while in the noisy setting, there is even no trivial upper bound on the sample complexity to achieve some desired approximation guarantee (e.g., our $(5, \epsilon)$-approximation). Note that the value of $\tilde{\Delta}_{e,\epsilon}$ defined in (2) always has the following lower bound: $\tilde{\Delta}_{e,\epsilon} = \Delta_e + \epsilon/2 \ (> 0)$ if $\epsilon/2 \geq \Delta_{\min}$ holds and $\tilde{\Delta}_{e,\epsilon} = \Delta_e + \epsilon - \Delta_{\min} \geq \epsilon \ (> 0)$ otherwise. Therefore, our sample complexity $T$ given in Theorem 1 is always bounded, contrasting existing results for threshold bandits [51]. The naive sampling algorithm (Uniform-FC in Appendix E) requires $O(\frac{n^6}{\epsilon^2} \log \frac{n}{\delta})$ samples to achieve the $(5, \epsilon)$-approximation w.p. at least $1 - \delta$. KC-FC achieves a much better sample complexity than Uniform-FC, as $\sum_{e \in E} \tilde{\Delta}_{e,\epsilon'}^{-2} = \sum_{e \in E}(\Delta_e + \frac{\epsilon'}{2})^{-2} \ll \frac{n^6}{\epsilon^2}$ when $\Delta_{\min} \leq \frac{\epsilon'}{2} \ll \Delta_e$ for most $e \in E$, which is often the case in practice. To the best of our knowledge, lower bounds on the sample complexity related to PE-CMAB are known only for the following settings: (i) the time complexity of algorithms can be exponential, or (ii) the underlying offline problem is assumed to be polynomial-time solvable and to have the unique correct (namely optimal) solution [25, 35, 39]. Deriving an effective lower bound on the number of samples required to guarantee an approximate solution is particularly challenging because it necessitates dealing with multiple correct solutions [33], while most existing approaches rely on the uniqueness of the correct solution. Evaluating the necessity of the second term $\frac{n^2}{\max\{\Delta_{\min}, \frac{\epsilon'}{2}\}^2}$ and investigating a lower bound for our case are crucial and remain important future work. However, it is worth noting that the additional term is independent of a dominating term involving $\log \frac{1}{\delta}$.

**Remark.** If we utilize TB-HS within the loop (Algorithm 5 in Appendix B), the algorithm achieves $(5, \epsilon)$-approximation guarantee with probability at least $1 - \delta$, and the sample complexity $T$ is:

$$
\mathcal{O}\left( \sum_{r=1}^{k} \left( \sum_{e \in I_{V_r}(p_r)} \frac{1}{\tilde{\Delta}_{e,\epsilon'_r}^2} \log\left( \frac{n}{\tilde{\Delta}_{e,\epsilon'_r}^2 \delta} \log\left( \frac{n}{\tilde{\Delta}_{e,\epsilon'_r}^2 \delta} \right) \right) + \frac{|V_r|}{\max(\Delta_{\min,r}, \frac{\epsilon'_r}{2})^2} \right) \right),
$$

where $k$ is the total number of loops in Algorithm 5, $\epsilon'_r := \epsilon/(12|I_{V_r}(p_r)|)$, $I_{V_r}(p_r) \subseteq E$ represents the set of pairs between the pivot $p_r$ selected in phase $r$ and its neighbors in $V_r$, and $\Delta_{\min,r} := \min_{e \in I_{V_r}(p_r)} \Delta_e$. When $k \ll n$, the above sample complexity can be better than that of Theorem 1. However, it should be noted that the symbols related to $r$ and the total number of loops $k$, especially instance-dependent gaps $\tilde{\Delta}_{e,\epsilon'_r}$, are all random variables. In contrast, the current Theorem 1 does not contain any random variables. Specifically, the significant term related to $\log \delta^{-1}$ is characterized by the gap $\tilde{\Delta}_{e,\epsilon}$ or $\Delta_e$, which represents the distance from 0.5 and not a random variable.

## 4 Fixed budget setting

In this section, we investigate Problem 2 and design KC-FB (Algorithm 3). KC-FB is inspired by the successive reject algorithm [6] and exploits KwikCluster to determine the number of queries for each pair adaptively.

**Algorithm.** KC-FB proceeds in at most $n$ phases and maintains the subset of elements $V_r \subseteq V$ in each phase $r \in [n]$ starting with $V_1 = V$. We denote the set of pairs that can be formed with $v$ in $V_r$ by $I_{V_r}(v) := \{\{v, u\} \in \binom{V_r}{2} : u \in V_r\}$. In each phase $r$, the algorithm chooses the pivot $p_r$ uniformly at random from $V_r$, and pulls each $e \in I_{V_r}(p_r)$ for appropriately determined $\tau_r$ times. Based on the empirical mean $\widehat{s}_r(e) := \sum_{k=1}^{\tau_r} X_k(e)/\tau_r$ for each $e \in I_{V_r}(p_r)$, it finds one cluster $C_r = \{p_r\} \cup \Gamma_{V_r}(p_r, \widehat{s}_r)$, where $\Gamma_{V_r}(p_r, \widehat{s}_r) := \{u \in V_r : \widehat{s}_r(p_r, u) > 0.5\}$, and updates $V_{r+1} \leftarrow V_r \setminus C_r$. This procedure will be continued until $|V_r| = 0$ and finally the algorithm outputs $\mathcal{C}_{\text{out}}$ consisting of all clusters computed. Updating the number of pulls $\tau_r$ (line 8) is a key to prove the statistical property. Intuitively, $\tau_r$ represents a pre-fixed budget of queries when $e \in \binom{V_r}{2}$ would be pulled: In the initial phase, we allocate $\tau_1 := \lfloor T/m \rfloor$ to each $e \in \binom{V_1}{2}$. Notice that the surplus, the sum of the pre-fixed budgets of pairs that have been removed without being queried, is $\tau_1 \cdot \left( |\binom{V_1}{2}| - |\binom{V_2}{2}| - (|V_1| - 1) \right)$, because the number of pairs that have been removed in this phase is $|\binom{V_1}{2}| - |\binom{V_2}{2}|$, and among those pairs, the number of pairs that have been actually pulled by the algorithm is $(|V_1| - 1)$. This surplus is additionally redistributed equally to each $e \in \binom{V_2}{2}$. This will be also done for the remaining phases $r = 2, \ldots, n$.

**Algorithm 3** KwikCluster with Fixed Budget (KC-FB)

---

**Input** : Budget $T > 0$, set $V$ of $n$ objects, additive error $\epsilon$

1   $V_1 \leftarrow V$, $r \leftarrow 1$, $\tau_1 \leftarrow \lfloor T/m \rfloor$, and $\mathcal{C}_{\text{out}} \leftarrow \emptyset$;

2   **while** $|V_r| > 0$ **do**

3      Pick a pivot $p_r \in V_r$ uniformly at random;

4      Pull each $e \in I_{V_r}(p_r)$ for $\tau_r$ times and observe random feedback $\{X_k(e)\}_{k=1}^{\tau_r}$;

5      Compute empirical mean $\widehat{s}_r(e) = \sum_{k=1}^{\tau_r} X_k(e)/\tau_r$ for each $e \in I_{V_r}(p_r)$;

6      $\mathcal{C}_{\text{out}} \leftarrow \mathcal{C}_{\text{out}} \cup \{C_r\}$ where $C_r := \{p_r\} \cup \Gamma_{V_r}(p_r, \widehat{s}_r)$;

7      $V_{r+1} \leftarrow V_r \setminus C_r$;

8      $\tau_{r+1} \leftarrow \tau_r + \left\lfloor \frac{\tau_r \cdot (|\binom{V_r}{2}| - |\binom{V_{r+1}}{2}| - (|V_r| - 1))}{|\binom{V_{r+1}}{2}|} \right\rfloor$ and $r \leftarrow r + 1$;

9   **return** $\mathcal{C}_{\text{out}}$

---

**Analysis.** The following theorem states that KC-FB outputs a well-approximate solution with high probability. The proof of Theorem 2 is deferred to Appendix D.

**Theorem 2.** *For $\epsilon > 0$, define the minimal gap $\Delta_{\min,\epsilon}$ as*

$$\min_{e \in E} \max \left\{ \frac{\epsilon}{6 \max\{1, |E_{[0.5 \pm \epsilon]}|\}}, \Delta_e \right\} \quad \text{for } \epsilon \in (0, 0.5),$$
$$\min_{e \in E} \max \left\{ \frac{\epsilon}{6m}, \Delta_e \right\} \qquad\qquad \text{for } \epsilon \geq 0.5,$$

*where $\Delta_e = |s(e) - 0.5| \ (\forall e \in E)$.*

*Then, KC-FB (Algorithm 3) uses at most $T$ queries to output $\mathcal{C}_{\text{out}}$ that satisfies*

$$\Pr[\mathbb{E}[\text{cost}_s(\mathcal{C}_{\text{out}})] \leq 5 \cdot \text{OPT}(s) + \epsilon] \geq 1 - \delta \ \text{ for } \delta \leq 2n^3 \exp\left( -\frac{2T\Delta_{\min,\epsilon}^2}{n^2} \right). \tag{3}$$

*Assuming that each query takes $\mathcal{O}(1)$ time, the time complexity of KC-FB is $\mathcal{O}(T + n^2)$.*

*Proof Sketch.* We can show the random event $\Pr\left[\bigcap_{r=1}^{n} \mathcal{E}_r\right]$ occurs with high probability, where $\mathcal{E}_r := \{\forall e \in I_{V_r}(p_r), \ |s(e) - \widehat{s}_r(e)| < \max\{\epsilon, \Delta_e\}\}$ for each phase $r \in [n]$ (See Lemma 8 in Appendix D.1). Under the assumption of such estimation success event $\bigcap_{r=1}^{n} \mathcal{E}_r$, by utilizing the unique feature of KwikCluster, we can maintain the approximation guarantee of KwikCluster in the noise-free setting up to additive error (See Lemma 9 in Appendix D.2). Simply combining these lemmas with adjusted parameter $\epsilon' \in (0, 0.5)$, defined as $\frac{\epsilon}{6 \max\{1, |E_{[0.5 \pm \epsilon]}|\}}$ if $\epsilon < 0.5$ and $\frac{\epsilon}{6m}$ otherwise, will conclude the proof (See Appendix D.3 for details). $\qquad \square$

The parameter $\delta \in (0, 1)$ in (3) represents the *error probability* of $\mathcal{C}_{\text{out}}$ being worse than any $(5, \epsilon)$-approximate solution, and it decays exponentially to the querying budget $T$. A larger parameter $\Delta_{\min,\epsilon}$ provides the better guarantee; KC-FB performs better when the similarity function clearly expresses similarity ($+1$) or dissimilarity ($-1$), as $\min_{e \in E} \Delta_e$ tends to be large.

To evaluate the significance of our results, we analyze the uniform sampling algorithm (Uniform-FB in Appendix E); Uniform-FB queries each $e \in E$ uniformly $\lfloor T/m \rfloor$ times to obtain $\widehat{s}(e)$, and then applies any $\alpha$-approximation algorithm to instance $(V, \widehat{s})$ of the offline problem minimizing (1). We see that the error probability that the output is not an $(\alpha, \epsilon)$-approximate solution is bounded by $\mathcal{O}\left( n^2 \exp\left( -\frac{T\epsilon^2}{\alpha^2 n^6} \right) \right)$. In contrast, KC-FB adaptively allocates the budget to the remaining pairs, which enables us to query essential pairs of elements, i.e., pairs whose estimated similarity values affect the behavior of cluster construction, more times than $\lfloor T/m \rfloor$. This leads to a better performance in the cost of clustering in practice (see Section 5).

**Comparison with existing PE-CMAB methods in the FB setting.** In the literature of PE-CMAB, the FB setting presents even more computational challenges and a scarcity of theoretical results. The current state-of-the-art algorithms [6, 25, 36] suffer from one or more of the following issues: (i) inability to handle a partition structure in correlation clustering, (ii) requiring exponential running time, and (iii) lacking any approximation guarantees when the underlying problem is NP-hard. By leveraging the properties of KwikCluster, our approach ensures the polynomial-time complexity of $O(T + n^2)$ while guaranteeing that the probability of obtaining a well-approximate solution exponentially increases with the budget $T$, along with instance-dependent analysis.

# 5 Experimental evaluation

In this section, we evaluate the performance of our proposed algorithms, KC-FC and KC-FB, using various datasets, providing empirical evidence to support our theoretical findings.

**Datasets.** We use publicly-available real-world graphs presented in Table 1. In the FC setting, to observe the behavior of the sample complexity with respect to the hidden minimum gap $\Delta_{\min}$ in (2), we generate our instances as follows. For each graph, we vary the lower bound on $\Delta_{\min}$, which we denote by $\mathrm{LB}_{\Delta_{\min}}$, in $\{0.10, 0.15, 0.20, \dots, 0.50\}$. For each pair

Table 1: Real-world graphs used in our experiments.

| Name | # of vertices | # of edges | Description |
|---|---|---|---|
| Lesmis | 77 | 254 | Co-appearance network |
| Adjnoun | 112 | 425 | Word-adjacency network |
| Football | 115 | 613 | Sports team network |
| Jazz | 198 | 2,742 | Social network |
| Email | 1,133 | 5,451 | Communication network |
| ego-Facebook | 4,039 | 88,234 | Social network |
| Wiki-Vote | 7,066 | 100,736 | Wikipedia voting network |

of vertices $u, v$, we set $\mathrm{s}(u, v) = \mathrm{uniform}[0.5 + \mathrm{LB}_{\Delta_{\min}}, 1]$ if $u, v$ have an edge in the graph, and $\mathrm{s}(u, v) = \mathrm{uniform}[0, 0.5 - \mathrm{LB}_{\Delta_{\min}}]$ otherwise, where $\mathrm{uniform}[a, b]$ is the value drawn from the interval $[a, b]$ uniformly at random. On the other hand, in the FB setting, we employ a more realistic setting: For each graph, our problem instance is generated by embedding the vertices into a $d$-dimensional Euclidean space using node2vec [43], obtaining a vector $\mathrm{vec}(v) \in \mathbb{R}^d$ for each vertex $v$. Specifically, we used the publicly-available Python module of node2vec[2] with default parameter settings (particularly $d = 64$). Then, define the unknown similarity of each pair of vertices $u, v$ as $\mathrm{s}(u, v) = \frac{\mathrm{sim}_{\cos}(\mathrm{vec}(u), \mathrm{vec}(v)) - \min\_\cos}{\max\_\cos - \min\_\cos} \in [0, 1]$, where $\min\_\cos$ and $\max\_\cos$ are the minimal and maximal cosine similarities, respectively, among all pairs of vertices. We note that $\max\_\cos > \min\_\cos$ holds for all instances. In all experiments, noisy feedback when querying a pair $e \in E$ is generated by a Bernoulli distribution with mean $\mathrm{s}(e)$.

**Baselines.** We compare our methods with Uniform-FC in the FC setting and Uniform-FB in the FB setting, whose pseudocode and full analysis are given in Appendix E. Uniform-FC pulls each $e \in E$ for $\lceil \frac{18m^2}{\epsilon^2} \log \frac{2m}{\delta} \rceil$ times and employs KwikCluster with respect to the empirical similarity, while Uniform-FB is its adaption to the FB setting. Moreover, we compare the cost of clustering of our algorithms with that of KwikCluster having access to the unknown (true) similarity, which is regarded as the stronger baseline than other KwikCluster-based methods for the binary case [11, 15, 40, 76].

**Machine and code.** The experiments were performed on a machine with Apple M1 Chip and 16 GB RAM. The code was written in Python 3, which is available online.[3]

**Performance of KC-FC.** We evaluate the performance of algorithms in terms of not only the cost of clustering but also the sample complexity. In both KC-FC and Uniform-FC, we set $\epsilon = \sqrt{n}$ allowing each element to make only $1/\sqrt{n}$ mistakes, and $\delta = 0.01$ following a standard choice in PE-MAB. Taking into account the limited scalability of the algorithms, we only use the instances with $n < 1,000$. In particular, as will be shown later, Uniform-FC requires a large number of samples, which makes the algorithm prohibitive even for quite small instances. Therefore, we do not run the algorithm and just report the sample complexity, which can be calculated without running it. For each $\mathrm{LB}_{\Delta_{\min}}$, we run both KC-FC and KwikCluster having access to the unknown similarity 100 times and report the average value and the standard deviation.

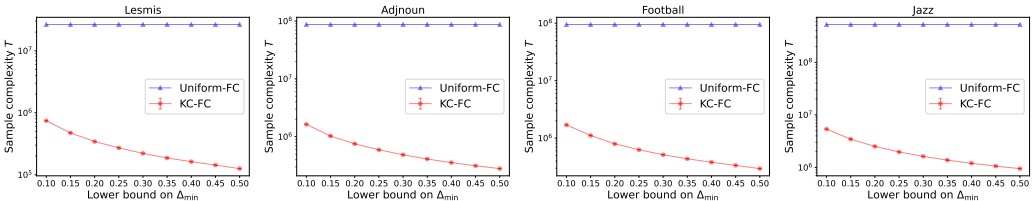

Figure 1: Sample complexity of KC-FC & Uniform-FC.

The results are depicted in Figures 1 and 2. As can be seen, the sample complexity of KC-FC is much smaller than that of Uniform-FC. In fact, the sample complexity of Uniform-FC makes the algorithm

[2] https://pypi.org/project/node2vec/
[3] https://github.com/atsushi-miyauchi/CC-Bandits

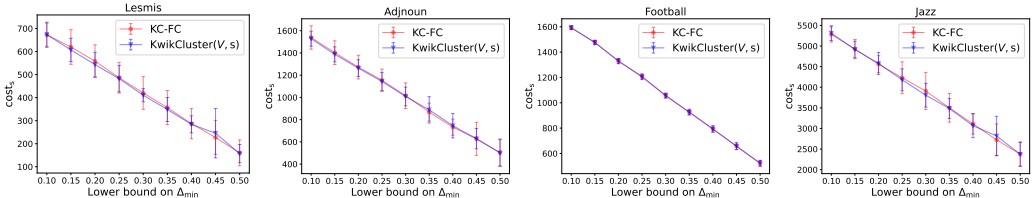

Figure 2: Cost of clustering of KC-FC & KwikCluster having the access to the unknown similarity.

prohibitive even for very small instances. Moreover, consistent with the theoretical analysis, as (the lower bound $\mathrm{LB}_{\Delta_{\min}}$ on) $\Delta_{\min}$ increases, the sample complexity of KC-FC becomes smaller. This desirable property is not possessed by Uniform-FC. Remarkably, looking at Figure 2, we see that KC-FC outputs a clustering whose quality is comparable with that of KwikCluster having access to the unknown similarity.

Table 2: Cost of clustering of KC-FB & baselines ($n \geq 1,000$).

| Name | KC-FB | Uniform-FB | KwikCluster($V, \mathrm{s}$) |
|---|---|---|---|
| Email | 218k±1.1k | 221k±0.5k | 209k±0.5k |
| ego-Facebook | 3,716k±36.5k | 3,780k±29.6k | 3,373k±59.8k |
| Wiki-Vote | 10,222k±45.5k | 10,428k±32.0k | 9,749k±34.7k |

**Performance of KC-FB.** Here we evaluate the performance of KC-FB. For small instances with $n < 1,000$, we vary $T$ in $\{n^{2.1}, n^{2.2}, \ldots, n^{3.0}\}$ and observe the cost of clustering with respect to the budget $T$. For large instances with $n \geq 1,000$, we fix $T = n^{2.2}$ for scalability. For each instance and $T$, we run both KC-FB and Uniform-FB 100 times and report the average value and the standard deviation. As KwikCluster having access to the unknown similarity is independent of $T$, we just run it 100 times for each instance.

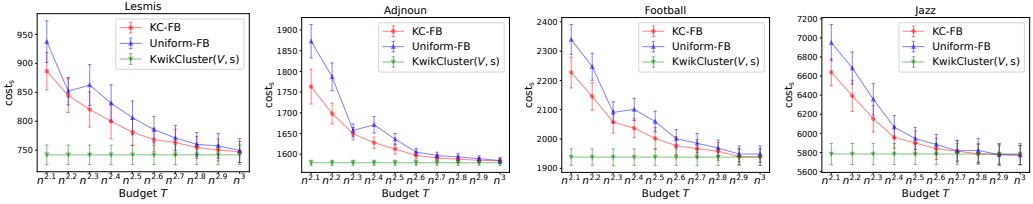

Figure 3: Cost of clustering of KC-FB & baselines ($n < 1,000$).

The results are shown in Figure 3 and Table 2. As can be seen, KC-FB outperforms the baseline method Uniform-FB. In fact, for all instances and almost all values of $T$, KC-FB outputs a better clustering than that of Uniform-FB. We can see that this superiority comes from the fact that KC-FB estimates the unknown similarity better than Uniform-FB thanks to its sophisticated sampling strategy. Indeed, KwikCluster having access to the unknown similarity showcases the best performance, verifying the importance of the precise estimation of the unknown similarity.

## 6 Conclusions

We studied the online learning problems of correlation clustering, where the similarity function is initially unknown and only noisy feedback is observed. For the FC setting, we devised KC-FC and proved the upper bound of the number of queries required to find a clustering whose cost is at most $5 \cdot \mathrm{OPT} + \epsilon$ with high probability. For the FB setting, we devised KC-FB and showed that the error probability of the expected cost being worse than $5 \cdot \mathrm{OPT} + \epsilon$ decays exponentially with budget $T$. Importantly, our algorithms are the first examples of PE-CMAB with NP-hard offline problems. One future work, yet a significant challenge, is to derive information-theoretic lower bounds of PE-CMAB in the case where the offline problem is NP-hard. Investigating other variants of correlation clustering or exploring the case where the variance of random feedback differs across pairs, namely heteroscedastic noise, would also be worthwhile directions.

## Acknowledgment

The work of Yuko Kuroki is supported by Japan Science and Technology Agency (JST) Strategic Basic Research Programs PRESTO "R&D Process Innovation by AI and Robotics: Technical Foundations and Practical Applications" grant number JPMJPR24T2, and was partially supported by Microsoft Research Asia and JST Strategic Basic Research Programs ACT-X grant number JPMJAX200E while she was at The University of Tokyo. The authors would like to thank the anonymous reviewers for their insightful comments and useful feedback.

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

# Appendix

## A   Additional comparison with other clustering models

**Cluster recovering with noisy same-cluster queries.**   Another line of studies has focused on clustering reconstruction with noisy same-cluster queries, which was first proposed by Mazumdar and Saha [65] and further investigated by Larsen et al. [58], Peng and Zhang [70], Pia et al. [71], Tsourakakis et al. [77] and Xia and Huang [83]. In this model, given a set of $n$ elements, the goal is to recover the underlying ground-truth $k$-clustering by asking pairwise queries to an oracle, which tells us if the two elements belong to the same cluster, but whose answer is correct only with probability $\frac{1}{2} + \frac{\delta}{2}$. Recently, Gupta et al. [45] first considered a noisy and inconsistent oracle, in contrast to a consistent oracle that returns the same answer when queried. Although this clustering reconstruction problem shares the intuition with correlation clustering, there are also important differences which do not allow to transfer algorithmic results from one to the other. Firstly, in correlation clustering the input information might be inconsistent (e.g., $a$ is very similar to $b$, which is very similar to $c$, but $a$ is not similar to $c$), instead in clustering reconstruction this is not possible: if $a$ is in the same cluster of $b$ and $b$ is in the same cluster of $c$, then $a$ is in the same cluster of $c$. Secondly, the aim is to reconstruct the exact underlying clustering, while we aim at minimizing the cost function in (1). Lastly and more importantly, the number of clusters $k$ is part of the input to the problem, while in correlation clustering it is unknown. Therefore, the theoretical results and techniques for solving the clustering reconstruction problem cannot be directly applied to correlation clustering.

**Query-based correlation clustering.**   As fully discussed in the main text, our work lies in query-efficient correlation clustering, for which we utilize the methodology of PE-CMAB. Table 3 summarizes how existing query-based settings differ from our attempt. Note that the oracle in Aronsson and Chehreghani [4, 5] is assumed to return the true value of s$(x, y)$ with probability $1 - \gamma$ and a noisy value with probability $\gamma$, which is different from our models; we only observe a random variable independently sampled from an unknown distribution with mean s$(x, y) \in [0, 1]$. While one might consider using majority voting on repeated queries to handle noise when the underlying distribution is Bernoulli, this approach lacks any approximation guarantees and query complexity bounds. Moreover, for $R$-sub-Gaussian noise, majority voting is not well-defined. Instead, using sample mean estimates, as done in PE-MAB methods, is standard. Our approach leverages these principles, ensuring a $(5, \epsilon)$-approximation guarantee with fewer queries and statistical guarantees.

Table 3: Different problem settings in correlation clustering with queries.

| Feature/Study | Similarity Function | Similarity Type | Oracle | Theoretical Guarantee |
|---|---|---|---|---|
| Ailon et al. [2] Saha and Subramanian [74] | Known | Binary | Strong with access to same-cluster queries in the optimal clustering | ✓ |
| Bonchi et al. [11] Bressan et al. [15] García-Soriano et al. [40] | Unknown | Binary | Strong with access to the true value of s$(x, y) \in \{0, 1\}$ | ✓ |
| Silwal et al. [76] | Unknown | Binary | Both strong with access to the true value of s$(x, y) \in \{0, 1\}$ and noisy | ✓ |
| Aronsson and Chehreghani [4, 5] | Unknown | Weighted | Noisy (true value of s$(x, y)$ is returned with probability $1 - \gamma$ and a noisy value is retuned otherwise) | Not provided |
| **Our Work** | **Unknown** | **Weighted** | **Noisy (stochastic feedback)** | ✓ |

## B   Pseudocode of **KwikCluster** and Algorithm 5

We detail the pseudocode of KwikCluster [3] in Algorithm 4. The approximation guarantee of KwikCluster is $5$ for the weighted similarity case (and $3$ for the restricted binary similarity case). We also detail the pseudocode of Algorithm 5, which sequentially uses TB-HS at each phase in the framework of KC-FC.

---

**Algorithm 4** KwikCluster$(V, \mathrm{s})$

---

**Input** : Set $V$ of $n$ objects, and similarity function s

1 $\mathcal{C} \leftarrow \emptyset$;
2 **while** $|V| > 0$ **do**
3      Pick a pivot $p \in V$ uniformly at random;
4      $\mathcal{C} \leftarrow \mathcal{C} \cup \{C_p\}$ where $C_p := \{p\} \cup \{u \in V : \mathrm{s}(p, u) > 0.5\}$;
5      $V \leftarrow V \setminus C_p$;
6 **return** $\mathcal{C}$

---

**Algorithm 5** KC-FC variant with sequential use of TB-HS

---

**Input** : Confidence level $\delta$, set $V$ of $n$ objects, and error $\epsilon$

7 $E_1 \leftarrow E, V_1 \leftarrow V, r \leftarrow 1$, and $\mathcal{C}_{\mathrm{out}} \leftarrow \emptyset$;
8 **while** $|V_r| > 0$ **do**
9      Pick a pivot $p_r \in V_r$ uniformly at random;
10      Let $I_{V_r}(p_r) \subseteq E$ be the set of pairs between the pivot $p_r$ and its neighbors in $V_r$;
11      $\epsilon'_r := \epsilon/(12|I_{V_r}(p_r)|)$;
12      Compute $\widehat{G}^{(r)}_{\epsilon'}$ by TB-HS (Algorithm 2) with the input of error $\epsilon'_r$, confidence level $\delta/n$, and the set of pairs $I_{V_r}(p_r)$;
13      Define $\widehat{\Gamma}^{(r)}(v) := \{u \in V : \{u, v\} \in \widehat{G}^{(r)}_{\epsilon'}\}$;
14      $\mathcal{C}_{\mathrm{out}} \leftarrow \mathcal{C}_{\mathrm{out}} \cup \{C_r\}$, where $C_r := (\{p_r\} \cup \widehat{\Gamma}^{(r)}(p_r)) \cap V_r$;
15      $V_{r+1} \leftarrow V_r \setminus C_r$ and $r \leftarrow r + 1$;
16 **return** $\mathcal{C}_{\mathrm{out}}$

---

## C  Analysis of KC-FC

In this section, we provide a complete proof of Theorem 1 in Section 3. In particular, Lemma 4 guarantees the accuracy of the subroutine TB-HS, and based on this, Lemma 5 assures the $(5, \epsilon)$-approximation using the properties of KwikCluster. Moreover, Lemma 6 establishes an upper bound crucial for the sample complexity via novel analysis dependent on $\epsilon$ and $\Delta_{\min}$ (Lemma 7). Finally, by combining the sample complexity required by subroutine TB-HS (Lemma 6) and the output guarantee of KC-FC (Lemma 5), Theorem 1 is demonstrated.

### C.1  Basic lemmas

We first introduce the Hoeffding inequality, which will be frequently used in our proof. Note that we consider Bernoulli distribution for the sake of simplicity, but our results would carry on for $R$-sub-Gaussian distribution by simply adjusting the Hoeffding inequality for the case accordingly.

**Lemma 1** (Hoeffding inequality for bounded random variables)**.** *Let $X_1, \ldots, X_k$ be $k$ independent random variables such that, $\mathbb{E}[X_i] = \mu$ and $a \leq X_i \leq b$ for each $i \in [k]$. Let $\bar{X} = \frac{1}{k} \sum_{i=1}^{k} X_i$ denote the average of these random variables. Then, for any $\lambda > 0$, we have*

$$\Pr\left[\bar{X} \leq \mu - \lambda\right] \leq \exp\left(-\frac{2k\lambda^2}{(b-a)^2}\right).$$

The next lemma presents the probability that some random event happens, which will be used later.

**Lemma 2.** *Let $\widehat{\mathrm{s}}_{e,k}$ be the empirical mean of the rewards when $e$ has been pulled $k$ times. For each $e \in [m]$ and $k$, define the random event $\mathcal{E}_{e,k}$ as follows:*

$$\mathcal{E}_{e,k} := \left\{|\mathrm{s}(e) - \widehat{\mathrm{s}}_{e,k}| < \sqrt{\frac{\log(4mk^2/\delta)}{2k}}\right\}.$$

*Let $\mathcal{E}_k$ be the random event that for all $e \in [m]$, the random event $\mathcal{E}_{e,k}$ happens. Then we have*

$$\Pr\left[\bigcap_{k=1}^{\infty} \mathcal{E}_k\right] \geq 1 - \delta.$$

*Proof of Lemma 2.* We have

$$\Pr\left[\bigcup_{k=1}^{\infty}\neg\mathcal{E}_{e,k}\right] = \sum_{k=1}^{\infty}\Pr\left[|s(e)-\widehat{s}_{e,k}|\geq\sqrt{\frac{\log(4mk^2/\delta)}{2k}}\right] \leq \sum_{k=1}^{\infty}\frac{\delta}{2mk^2} = \frac{\pi^2\delta}{12m} \leq \frac{\delta}{m},$$

where the first inequality follows from the Hoeffding inequality (Lemma 1). Therefore, by taking the union bound, we have

$$\Pr\left[\bigcap_{k=1}^{\infty}\mathcal{E}_k\right] \geq 1 - \Pr\left[\bigcup_{e\in[m]}\bigcup_{k=1}^{\infty}\neg\mathcal{E}_{e,k}\right] \geq 1 - \delta.$$

$\square$

## C.2  Approximation guarantee

We provide the following lemmas for guaranteeing the quality of the output.

**Lemma 3.** *Let $\epsilon \in (0, 0.5)$ and $\delta \in (0, 1)$. TB-HS (Algorithm 2) with parameter $\epsilon, \delta$ outputs, with probability at least $1 - \delta$, $\widehat{G}_\epsilon$ and $\widehat{B}_\epsilon$ such that*

$$s(e) \geq 0.5 - \epsilon \text{ for every } e \in \widehat{G}_\epsilon,$$
$$s(e) \leq 0.5 + \epsilon \text{ for every } e \in \widehat{B}_\epsilon,$$
$$\widehat{G}_\epsilon \cup \widehat{B}_\epsilon = E, \ \widehat{G}_\epsilon \cap \widehat{B}_\epsilon = \emptyset.$$

*Proof of Lemma 3.* The proof is almost straightforward by the procedure of the algorithm, line 8 and 10, as follows. By Lemma 2, we have $\Pr\left[\bigcap_{k=1}^{\infty}\mathcal{E}_k\right] \geq 1 - \delta$. Now we assume that $\bigcap_{k=1}^{\infty}\mathcal{E}_k$ happens. Let $t > 0$ be the stopping time, where every $e \in E$ has been added to either $\widehat{G}_\epsilon$ or $\widehat{B}_\epsilon$. For $e \in \widehat{G}_\epsilon$, from the stopping condition and the random event $\bigcap_{k=1}^{\infty}\mathcal{E}_k$, it is easy to see that $s(e) \geq \widehat{s}_{t'}(e) - \text{rad}_{t'}(e) \geq 0.5 - \epsilon$, where $t'$ denotes the round that arm $e$ was added to $\widehat{G}_\epsilon$. For $e \in \widehat{B}_\epsilon$, it is also easy to see that $s(e) \leq \widehat{s}_{t'}(e) + \text{rad}_{t'}(e) \leq 0.5 + \epsilon$, where $t'$ denotes the round that arm $e$ was added to $\widehat{B}_\epsilon$. The third condition is obvious from the stopping condition of the algorithm. $\square$

**Lemma 4.** *Let $\epsilon \in (0, 0.5)$ and $\delta \in (0, 1)$. Let $\widehat{G}_\epsilon$ and $\widehat{B}_\epsilon$ be the output of TB-HS (Algorithm 2) with parameters $\epsilon, \delta$. Then, with probability at least $1 - \delta$, we have that (i) every $e \in E_{(0.5+\epsilon,1]}$ is included in $\widehat{G}_\epsilon$, and (ii) every $e \in E_{[0,0.5-\epsilon)}$ is included in $\widehat{B}_\epsilon$.*

*Proof of Lemma 4.* We have $\Pr\left[\bigcap_{k=1}^{\infty}\mathcal{E}_k\right] \geq 1 - \delta$ by Lemma 2 again, and we assume that $\bigcap_{k=1}^{\infty}\mathcal{E}_k$ happens. Consider any $e \in E_{(0.5+\epsilon,1]}$. Suppose that $e$ is not included in $\widehat{G}_\epsilon$. Then, from Lemma 3, we see that $e \in \widehat{B}_\epsilon$, and thus $s(e) \leq 0.5 + \epsilon$, which contradicts the fact that $e \in E_{(0.5+\epsilon,1]}$. Therefore, $e$ is included in $\widehat{G}_\epsilon$. Similarly, consider any $e \in E_{[0,0.5-\epsilon)}$. Suppose that $e$ is not included in $\widehat{B}_\epsilon$. Then, from Lemma 3, we see that $e \in \widehat{G}_\epsilon$, and thus $s(e) \geq 0.5 - \epsilon$, which contradicts the fact that $e \in E_{[0,0.5-\epsilon)}$. Therefore, $e$ is included in $\widehat{B}_\epsilon$.

$\square$

Based on Lemma 4, we prove the following key lemma.

**Lemma 5** (Approximation guarantee)**.** *Let $\epsilon \in (0, 0.5)$ and $\delta \in (0, 1)$. With probability at least $1 - \delta$, the output $\mathcal{C}_{\text{out}}$ of KC-FC (Algorithm 1), where subroutine TB-HS (Algorithm 2) is invoked with parameters $\epsilon, \delta$, is a $(5, 12\epsilon|E_{[0.5\pm\epsilon]}|)$-approximate solution for instance $(V, s)$ of the offline problem minimizing (1).*

*Proof of Lemma 5.* By Lemma 4, we have $E_{(0.5+\epsilon,1]} \subseteq \widehat{G}_\epsilon$ and $E_{[0,0.5-\epsilon)} \subseteq \widehat{B}_\epsilon$ w.p. at least $1 - \delta$. Construct the similarity function $\widetilde{s} : E \to [0, 1]$ such that for each $e \in E$,

$$\widetilde{s}(e) = \begin{cases} s(e) & \text{if } e \in E_{[0,0.5-\epsilon)} \cup E_{(0.5+\epsilon,1]}, \\ \widetilde{s}_e & \text{otherwise,} \end{cases}$$

where $\widetilde{s}_e$ is an arbitrary value that satisfies $|s(e) - \widetilde{s}_e| < 2\epsilon$. Consider running KwikCluster with the similarity $\widetilde{s}$. Let $\mathcal{C}'_{\text{out}}$ be the output of this algorithm. Then we have

$$\mathbb{E}[\text{cost}_s(\mathcal{C}'_{\text{out}})] < \mathbb{E}[\text{cost}_{\widetilde{s}}(\mathcal{C}'_{\text{out}})] + 2\epsilon|E_{[0.5\pm\epsilon]}|$$
$$\leq 5 \cdot \text{OPT}(\widetilde{s}) + 2\epsilon|E_{[0.5\pm\epsilon]}|$$
$$< 5\left(\text{OPT}(s) + 2\epsilon|E_{[0.5\pm\epsilon]}|\right) + 2\epsilon|E_{[0.5\pm\epsilon]}|$$
$$= 5 \cdot \text{OPT}(s) + 12\epsilon|E_{[0.5\pm\epsilon]}|.$$

Noticing that KC-FC corresponds to the above algorithm associated with a certain choice of $\widetilde{s}$ (i.e., $\widetilde{s}_e$ for $e \in E_{[0.5\pm\epsilon]}$), we have the lemma.

$\square$

### C.3 Sample complexity analysis and proof of Theorem 1

We prove the following main lemma to evaluate the sample complexity of TB-HS. Let $m_{\text{g}}$ be the number of pairs (i.e., arms) whose similarity is no less than the threshold $0.5$. Without loss of generality, we assume that $E = [m]$ indexed as $s(1) \geq \cdots \geq s(m_{\text{g}}) \geq 0.5 > s(m_{\text{g}} + 1) \geq \cdots \geq s(m)$ in whole analysis.

**Lemma 6** (Sample complexity). *The upper bound of the sample complexity of TB-HS (Algorithm 2) with parameters $\epsilon \in (0, 0.5)$ and $\delta \in (0, 1)$ is*

$$T = \mathcal{O}\left(\sum_{e \in E} \frac{1}{\tilde{\Delta}_{e,\epsilon}^2} \log\left(\frac{\sqrt{m/\delta}}{\tilde{\Delta}_{e,\epsilon}^2} \log\left(\frac{\sqrt{m/\delta}}{\tilde{\Delta}_{e,\epsilon}^2}\right)\right) + \frac{m}{\max\{\Delta_{\min}, \epsilon/2\}^2}\right).$$

To prove Lemma 6, we begin with the following lemma. Recall that $\tilde{\Delta}_{e,\epsilon}$ and $\Delta_{\min}$ are defined by (2).

**Lemma 7.** *Let $\epsilon \in (0, 0.5)$ and $\delta \in (0, 1)$. Define*

$$k_e := \frac{1}{\tilde{\Delta}_{e,\epsilon}^2} \log\left(\frac{4\sqrt{m/\delta}}{\tilde{\Delta}_{e,\epsilon}^2} \log\left(\frac{5\sqrt{m/\delta}}{\tilde{\Delta}_{e,\epsilon}^2}\right)\right). \tag{4}$$

*Let $\underline{s}_{e,k} := \widehat{s}_{e,k} - \sqrt{\frac{\log(4mk^2/\delta)}{2k}}$, and $\bar{s}_{e,k} := \widehat{s}_{e,k} + \sqrt{\frac{\log(4mk^2/\delta)}{2k}}$, where $\widehat{s}_{e,k}$ is the empirical mean of the rewards when $e$ has been pulled $k$ times. If $k \geq k_e$ holds, then*

$$\Pr[\underline{s}_{e,k} \leq 0.5 - \epsilon] \leq \exp(-2k \max\{\Delta_{\min}, \epsilon/2\}^2), \ \forall e \in [m_g],$$
$$\Pr[\bar{s}_{e,k} \geq 0.5 + \epsilon] \leq \exp(-2k \max\{\Delta_{\min}, \epsilon/2\}^2), \ \forall e \in [m] \setminus [m_g].$$

*It also holds that*

$$\mathbb{E}\left[\sum_{k=1}^{\infty} \mathbb{1}\left[\underline{s}_{e,k} \leq 0.5 - \epsilon\right]\right] \leq k_e + \frac{1}{2\max\{\Delta_{\min}, \epsilon/2\}^2}, \ \forall e \in [m_g],$$
$$\mathbb{E}\left[\sum_{k=1}^{\infty} \mathbb{1}[\bar{s}_{e,k} \geq 0.5 + \epsilon]\right] \leq k_e + \frac{1}{2\max\{\Delta_{\min}, \epsilon/2\}^2}, \ \forall e \in [m] \setminus [m_g].$$

*Proof of Lemma 7.* Suppose that

$$\sqrt{\frac{\log(4mk^2/\delta)}{2k}} \leq \Delta_e - \max\left\{\Delta_{\min} - \epsilon, -\frac{\epsilon}{2}\right\}.$$

Then, for each $e \in [m_g]$, we have

$$\Pr[\underline{s}_{e,k} \leq 0.5 - \epsilon] = \Pr\left[\widehat{s}_{e,k} - s(e) \leq -\Delta_e - \epsilon + \sqrt{\frac{\log(4mk^2/\delta)}{2k}}\right]$$
$$\leq \Pr\left[\widehat{s}_{e,k} - s(e) \leq -\Delta_e - \epsilon + \Delta_e - \max\left\{\Delta_{\min} - \epsilon, -\frac{\epsilon}{2}\right\}\right]$$
$$= \Pr\left[\widehat{s}_{e,k} - s(e) \leq -\max\left\{\Delta_{\min}, \frac{\epsilon}{2}\right\}\right]$$
$$\leq \exp\left(-2k \max\left\{\Delta_{\min}, \frac{\epsilon}{2}\right\}^2\right),$$

where the last inequality follows from the Hoeffding equality (Lemma 1). Now we show, via a similar analysis of Lemma 2 in Kano et al. [51], that for $k \geq k_e$, it indeed holds that

$$\sqrt{\frac{\log(4mk^2/\delta)}{2k}} \leq \Delta_e - \max\left\{\Delta_{\min} - \epsilon, \ -\frac{\epsilon}{2}\right\}. \tag{5}$$

Let $c_e := \left(\Delta_e + \min\left\{\epsilon - \Delta_{\min}, \frac{\epsilon}{2}\right\}\right)^2$ for simplicity, Then we can rewrite $k \geq k_e$ as

$$k = \frac{1}{c_e}\log\frac{4t\sqrt{m/\delta}}{c_e}$$

for some $t \geq \log\frac{5\sqrt{m/\delta}}{c_e} > 1$. Then we have

$$\sqrt{\frac{\log(4mk^2/\delta)}{2k}} \leq \Delta_e + \min\left\{\epsilon - \Delta_{\min}, \ \frac{\epsilon}{2}\right\}$$

$$\Leftrightarrow \log(4mk^2/\delta) \leq 2c_e k$$

$$\Leftrightarrow \log\left(\frac{4m\left(\log\left(\frac{4t\sqrt{m/\delta}}{c_e}\right)\right)^2}{c_e^2\delta}\right) \leq \log\left(\frac{16t^2 m}{c_e^2\delta}\right)$$

$$\Leftrightarrow \log\left(\frac{4t\sqrt{m/\delta}}{c_e}\right) \leq 2t$$

$$\Leftarrow t - 1 + \log\left(\frac{4\sqrt{m/\delta}}{c_e}\right) \leq 2t$$

$$\Leftrightarrow \log\left(\frac{4\sqrt{m/\delta}}{\mathrm{e}\cdot c_e}\right) \leq t,$$

where e is the base of natural logarithms and $\log t \leq t - 1$ is used. Therefore, $t \geq \log\frac{5\sqrt{m/\delta}}{c_e}$ is sufficient to fulfill (5).

The second statement of Lemma 7 can easily be shown by adapting the proof of Lemma 3 in Kano et al. [51]. For each $e \in [m_g]$, we have

$$\mathbb{E}\left[\sum_{k=1}^{\infty}\mathbf{1}\left[\underline{\mathbf{S}}_{e,k} \leq 0.5 - \epsilon\right]\right] \leq \mathbb{E}\left[\sum_{k=1}^{k_e} 1 + \sum_{k=k_e+1}^{\infty}\mathbb{1}\left[\underline{\mathbf{S}}_{e,k} \leq 0.5 - \epsilon\right]\right]$$

$$\leq k_e + \sum_{k=1}^{\infty}\Pr[\underline{\mathbf{S}}_{e,k} \leq 0.5 - \epsilon]$$

$$\leq k_e + \sum_{k=1}^{\infty}\exp(-2k\max\{\Delta_{\min}, \epsilon/2\}^2)$$

$$\leq k_e + \frac{1}{\mathrm{e}^{2\max\{\Delta_{\min},\epsilon/2\}^2} - 1}$$

$$\leq k_e + \frac{1}{2\max\{\Delta_{\min}, \epsilon/2\}^2}.$$

For $e \in [m] \setminus [m_g]$, we omit the proof, as the analysis is essentially the same as the case for $e \in [m_g]$. $\qquad\square$

*Proof of Lemma 6.* Let $a(t) \in \binom{V}{2}$ denote the selected pair (i.e., arm) by the algorithm in round $t$. Then we have

$$
\begin{aligned}
T &= \sum_{t=1}^{\infty} \mathbb{1}[a(t) \in [m], t \leq T] \\
&= \sum_{t=1}^{\infty} \mathbb{1}[a(t) \in [m_g], t \leq T] + \sum_{t=1}^{\infty} \mathbb{1}[a(t) \in [m] \setminus [m_g], t \leq T] \\
&\leq \sum_{t=1}^{\infty} \mathbb{1}[a(t) \in [m_g]] + \sum_{t=1}^{\infty} \mathbb{1}[a(t) \in [m] \setminus [m_g]] \\
&\leq \sum_{e \in [m_g]} \sum_{t=1}^{\infty} \mathbb{1}[a(t) = e] + \sum_{e \in [m] \setminus [m_g]} \sum_{t=1}^{\infty} \mathbb{1}[a(t) = e] \\
&= \sum_{e \in [m_g]} \sum_{t=1}^{\infty} \sum_{k=1}^{\infty} \mathbb{1}[a(t) = e, N_t(e) = k] + \sum_{e \in [m] \setminus [m_g]} \sum_{t=1}^{\infty} \sum_{k=1}^{\infty} \mathbb{1}[a(t) = e, N_t(e) = k] \\
&\leq \sum_{e \in [m_g]} \sum_{k=1}^{\infty} \mathbb{1}\left[\bigcup_{t=1}^{\infty}\{a(t) = e, N_t(e) = k\}\right] + \sum_{e \in [m] \setminus [m_g]} \sum_{k=1}^{\infty} \mathbb{1}\left[\bigcup_{t=1}^{\infty}\{a(t) = e, N_t(e) = k\}\right],
\end{aligned}
$$

where the third inequality follows from the fact that event $\{a(t) = e, N_t(e) = k\}$ occurs for at most one $t \in \mathbb{N}$. For $e \in [m_g]$, we have

$$
\sum_{k=1}^{\infty} \mathbb{1}\left[\bigcup_{t=1}^{\infty}\{a(t) = e, N_t(e) = k\}\right] \leq \mathbb{E}\left[\sum_{k=1}^{\infty} \mathbb{1}\left[\mathrm{s}_{e,k} \leq 0.5 - \epsilon\right]\right] \leq k_e + \frac{1}{2\max\{\Delta_{\min}, \epsilon/2\}^2},
$$

where the second inequality follows from Lemma 7.

Similarly, for each $e \in [m] \setminus [m_g]$, we have

$$
\sum_{k=1}^{\infty} \mathbb{1}\left[\bigcup_{t=1}^{\infty}\{a(t) = e, N_t(e) = k\}\right] \leq \mathbb{E}\left[\sum_{k=1}^{\infty} \mathbb{1}[\bar{\mathrm{s}}_{e,k} \geq 0.5 + \epsilon]\right] \leq k_e + \frac{1}{2\max\{\Delta_{\min}, \epsilon/2\}^2}.
$$

Therefore, by combining the above, we have

$$
T \leq \sum_{e \in [m]} k_e + \frac{m}{2\max\{\Delta_{\min}, \epsilon/2\}^2},
$$

which concludes the proof. □

*Proof of Theorem 1.* Finally, we are ready to complete the proof of Theorem 1. In KC-FC, TB-HS is run with parameter $\epsilon' = \frac{\epsilon}{12m}$ and confidence $\delta$. Therefore, by Lemma 5 for $\epsilon', \delta$, we have the approximation guarantee:

$$
\mathbb{E}[\mathrm{cost}_{\mathrm{s}}(\mathcal{C}_{\mathrm{out}})] \leq 5 \cdot \mathrm{OPT}(\mathrm{s}) + 12\epsilon'|E_{[0.5\pm\epsilon']}| \leq 5 \cdot \mathrm{OPT}(\mathrm{s}) + \epsilon.
$$

The sample complexity of KC-FC is equal to that of TB-HS with parameters $\epsilon', \delta$, which is given by Lemma 6 for $\epsilon', \delta$. □

## D    Analysis of KC-FB

In this section, we prove Theorem 2 in Section 4.

### D.1    Basic analysis of some random event and its occurrence probability

The following lemma states that $\widehat{\mathrm{s}}_r$ for phase $r \in [n]$ is well-estimated with high probability. The proof is almost straightforward from the Hoeffding inequality (Lemma 1) and union bounds.

**Lemma 8.** *Let $\epsilon \in (0, 0.5)$. Given a phase $r \in [n]$, we define the random event*

$$\mathcal{E}_r := \{\forall e \in I_{V_r}(p_r), |\mathrm{s}(e) - \widehat{\mathrm{s}}_r(e)| < \max\{\epsilon, \Delta_e\}\}. \tag{6}$$

*Then, we have*

$$\Pr\left[\bigcap_{r=1}^{n} \mathcal{E}_r\right] \geq 1 - 2n^3 \exp\left(-\frac{2T \min_{e \in E} \max\{\epsilon, \Delta_e\}^2}{n^2}\right).$$

*Proof.* We first evaluate $\Pr\left[|\widehat{\mathrm{s}}_r(e) - \mathrm{s}(e)| \geq \max\{\epsilon, \Delta_e\}\right]$ for a fixed phase $r \in [n]$ and $e \in I_{V_r}(p_r)$. Each $e \in I_{V_r}(p_r)$ has been pulled at least $\lfloor T/m \rfloor$ times because the initial budget for the pair was set to $\tau_1 = \lfloor T/m \rfloor$ and the budget has not decreased in the later iterations. Then we have

$$\Pr\left[|\widehat{\mathrm{s}}_r(e) - \mathrm{s}(e)| \geq \max\{\epsilon, \Delta_e\}\right] = \Pr\left[\left|\sum_{k=1}^{\tau_r} X_k(e)/\tau_r - \mathrm{s}(e)\right| \geq \max\{\epsilon, \Delta_e\}\right]$$

$$\leq \Pr\left[\left|\sum_{k=1}^{\tau_r} X_k(e)/\tau_r - \mathrm{s}(e)\right| \geq \max\{\epsilon, \Delta_e\}\right]$$

$$\leq 2\exp\left(-2\tau_r \max\{\epsilon, \Delta_e\}^2\right)$$

$$\leq 2\exp\left(-\frac{2T \max\{\epsilon, \Delta_e\}^2}{m}\right), \tag{7}$$

where the second inequality follows from Lemma 1. Taking a union bound for $r \in [n]$ and all $e \in I_{V_r}(p_r)$, we further have

$$\Pr\left[\bigcap_{r=1}^{n} \mathcal{E}_r\right] \geq 1 - \sum_{r=1}^{n} \sum_{e \in I_{V_r}(p_r)} \Pr\left[|\widehat{\mathrm{s}}_r(e) - \mathrm{s}(e)| \geq \max\{\epsilon, \Delta_e\}\right]$$

$$\geq 1 - \sum_{r=1}^{n} \sum_{v \in V_r} \sum_{e \in I_{V_r}(v)} \Pr\left[|\widehat{\mathrm{s}}_r(e) - \mathrm{s}(e)| \geq \max\{\epsilon, \Delta_e\}\right]$$

$$\geq 1 - \sum_{r=1}^{n} \sum_{v \in V_r} \sum_{e \in I_{V_r}(v)} 2\exp\left(-\frac{2T \max\{\epsilon, \Delta_e\}^2}{m}\right)$$

$$= 1 - \sum_{r=1}^{n} 2|V_r||I_{V_r}(p_r)| \exp\left(-\frac{2T \min_{e \in E} \max\{\epsilon, \Delta_e\}^2}{m}\right)$$

$$= 1 - \sum_{r=1}^{n} 2|V_r|(|V_r| - 1) \exp\left(-\frac{2T \min_{e \in E} \max\{\epsilon, \Delta_e\}^2}{m}\right)$$

$$\geq 1 - \sum_{r=1}^{n} 2n^2 \exp\left(-\frac{2T \min_{e \in E} \max\{\epsilon, \Delta_e\}^2}{m}\right)$$

$$= 1 - 2n^3 \exp\left(-\frac{2T \min_{e \in E} \max\{\epsilon, \Delta_e\}^2}{m}\right),$$

where the third inequality follows from (7). $\qquad \square$

### D.2 Theoretical guarantee of the output

Next we prove a key lemma that provides the theoretical guarantee of the output $\mathcal{C}_{\mathrm{out}}$ of KC-FB.

**Lemma 9.** *Let $\epsilon \in (0, 0.5)$. Under the assumption that $\bigcap_{r=1}^{n} \mathcal{E}_r$ happens, the output $\mathcal{C}_{\mathrm{out}}$ of KC-FB is a $(5, 6\epsilon|E_{[0.5\pm\epsilon]}|)$-approximate solution for instance $(V, \mathrm{s})$ of the offline problem minimizing (1).*

*Proof.* Let $\widehat{E} \subseteq E$ be the set of pairs that have been pulled in the algorithm. For $e = \{u, v\} \in \widehat{E}$, let $r_e$ be the phase, in which either $u$ or $v$ is selected as a pivot. Construct the weight $\widetilde{\mathrm{s}} : E \to [0, 1]$ such

that for each $e \in E$,

$$\widetilde{s}(e) = \begin{cases} \widehat{s}_{r_e}(e) & \text{if } e \in E_{[0.5\pm\epsilon]} \cap \widehat{E}, \\ s(e) & \text{otherwise.} \end{cases}$$

Consider running KwikCluster (Algorithm 4) with the similarity $\widetilde{s}$ while respecting the selection of pivots $p_r$ of KC-FB, that is, in the $r$-th iteration, the algorithm selects the pivot $p_r$ if it exists. In the first iteration, the algorithm can select the pivot $p_1$ and construct the cluster $\{p_1\} \cup \Gamma_{V_1}(p_1, \widetilde{s})$. By the definition of $\widetilde{s}$ and the assumption of the lemma, we have $\Gamma_{V_1}(p_1, \widetilde{s}) = \Gamma_{V_1}(p_1, \widehat{s}_1)$. In fact, for any element $u$ in $V_1$ (except for $p_1$), we see that $\widetilde{s}(p_1, u) > 0.5$ if and only if $\widehat{s}(p_1, u) > 0.5$. Therefore, the cluster produced is exactly the same as $C_1$ in KC-FB. In the second iteration, the algorithm can select $p_2$ because $p_2$ was not contained in the cluster of the first iteration, and by applying the same argument as above, we see that the cluster of this iteration is exactly the same as $C_2$ in KC-FB. The later iterations can be handled in the same way. Therefore, we see that the output of the above algorithm coincides with that of KC-FB.

Then it suffices to show that the output of the above algorithm has the desired approximation guarantee. Let $\mathcal{C}'_{\text{out}}$ be the output of the above algorithm. Recalling that KC-FB picks pivot $p_t$ uniformly at random, we have

$$\begin{aligned} \mathbb{E}[\text{cost}_s(\mathcal{C}'_{\text{out}})] &\le \mathbb{E}[\text{cost}_{\widetilde{s}}(\mathcal{C}'_{\text{out}})] + \epsilon|E_{[0.5\pm\epsilon]}| \\ &\le 5 \cdot \text{OPT}(\widetilde{s}) + \epsilon|E_{[0.5\pm\epsilon]}| \\ &\le 5\left(\text{OPT}(s) + \epsilon|E_{[0.5\pm\epsilon]}|\right) + \epsilon|E_{[0.5\pm\epsilon]}| \\ &= 5 \cdot \text{OPT}(s) + 6\epsilon|E_{[0.5\pm\epsilon]}|, \end{aligned}$$

where the first and third inequalities follow from the fact that $s(e)$ and $\widetilde{s}(e)$ may be different only for $e \in E_{[0.5\pm\epsilon]}$ $(\cap \widehat{E})$ and the difference there is at most $\epsilon$ from the assumption of the lemma. $\quad\square$

### D.3 Proof of Theorem 2

*Proof of Theorem 2.* For $\epsilon > 0$, define $\epsilon' \in (0, 0.5)$ as $\frac{\epsilon}{6\max\{1, |E_{[0.5\pm\epsilon]}|\}}$ if $\epsilon < 0.5$ and $\frac{\epsilon}{6m}$ otherwise. By Lemma 8 for $\epsilon'$, we have that

$$\Pr\left[\bigcap_{r=1}^{n} \mathcal{E}'_r\right] \ge 1 - 2n^3 \exp\left(-\frac{2T\min_{e\in E}\max\{\epsilon', \Delta_e\}^2}{n^2}\right),$$

where $\mathcal{E}'_r$ is the random event for phase $r \in \{1, \ldots, n\}$ that is defined by (6) with $\epsilon'$. Therefore, using Lemma 9 for $\epsilon'$, we can see that the output $\mathcal{C}_{\text{out}}$ of KC-FB is a $(5, \epsilon)$-approximate solution for instance $(V, s)$ of the offline problem minimizing (1) w.p. at least $1 - 2n^3 \exp\left(-\frac{2T\min_{e\in E}\max\{\epsilon', \Delta_e\}^2}{n^2}\right)$.

Finally, we can easily confirm that KC-FB does not exceed the given budget $T$ due to the algorithm procedure of line 8, which concludes the proof.

$\square$

## E  Uniform sampling algorithms

Here we provide the complete description and analysis of the naive uniform-sampling algorithms for both the FC setting (Algorithm 6) and the FB setting (Algorithm 7). Note that in the FC setting, no feasible stopping conditions are known from previous studies to guarantee that the output is an approximate solution, even with uniform or arbitrary sampling strategies. Therefore existing analysis of uniform sampling given in Chen et al. [25] is not applicable to our case with offline optimization being NP-hard.

First, we show a basic analysis of the cost of clustering when the estimate $\widehat{s}$ is close to the unknown similarity $s$.

**Lemma 10.** *Let $\epsilon \in (0, 0.5)$. Assume that $|s(e) - \widehat{s}(e)| \le \epsilon$ for every $e \in E$. Let $\mathcal{C}_{\text{out}}$ be the output of any $\alpha$-approximation algorithm for instance $(V, \widehat{s})$ of the offline problem minimizing (1). Then $\mathcal{C}_{\text{out}}$ is an $(\alpha, (\alpha+1)\epsilon m)$-approximate solution for instance $(V, s)$ of the offline problem.*

---
**Algorithm 6** Uniform sampling in the FC setting (Uniform-FC)
---
**Input**  : Set $V$ of $n$ objects, confidence level $\delta$, additive error $\epsilon > 0$
1  $T(e) \leftarrow \lceil \frac{(\alpha+1)^2 m^2}{2\epsilon^2} \log \frac{2m}{\delta} \rceil$ for each $e \in E$;
2  Sample each $e \in E$ for $T(e)$ times and compute empirical mean $\widehat{s}(e)$;
3  $\hat{\mathcal{C}} \leftarrow$ solution of an approximation algorithm for instance $(V, \widehat{s})$ of the offline problem minimizing (1);
4  **return** $\mathcal{C}_{\text{out}} := \hat{\mathcal{C}}$
---

---
**Algorithm 7** Uniform sampling in the FB setting (Uniform-FB)
---
**Input**  : Set $V$ of $n$ objects, budget $T$
1  Sample each $e \in E$ for $\lfloor T/m \rfloor$ times and compute the empirical mean $\widehat{s}(e)$;
2  $\hat{\mathcal{C}} \leftarrow$ solution of an approximation algorithm for instance $(V, \widehat{s})$ of the offline problem minimizing (1);
3  **return** $\mathcal{C}_{\text{out}} := \hat{\mathcal{C}}$
---

*Proof.* We have

$$\begin{aligned}
\mathbb{E}[\text{cost}_s(\mathcal{C}_{\text{out}})] &\leq \mathbb{E}[\text{cost}_{\widehat{s}}(\mathcal{C}_{\text{out}})] + \epsilon m \\
&\leq \alpha \cdot \text{OPT}(\widehat{s}) + \epsilon m \\
&\leq \alpha \left(\text{OPT}(s) + \epsilon m\right) + \epsilon m \\
&= \alpha \cdot \text{OPT}(s) + (\alpha + 1)\epsilon m.
\end{aligned}$$

$\square$

Next we evaluate the performance of Algorithm 6.

**Proposition 1.** *Given a confidence level $\delta \in (0, 1)$ and an additive error $\epsilon \in (0, 0.5)$, the uniform sampling algorithm with an $\alpha$-approximation oracle for the FC setting (Algorithm 6) outputs $\mathcal{C}_{\text{out}}$ that satisfies*

$$\Pr\left[\text{cost}_s(\mathcal{C}_{\text{out}}) \leq \alpha \cdot \text{OPT}(s) + \epsilon\right] \geq 1 - \delta,$$

*and the upper bound of the number of samples is*

$$T = \mathcal{O}\left(\frac{\alpha^2 n^6}{\epsilon^2} \log \frac{2n^2}{\delta}\right).$$

*Proof.* As the algorithm samples each $e \in E$ for $T(e)$ times, by the Hoeffding inequality (Lemma 1), it holds that

$$\Pr\left[|\widehat{s}(e) - s(e)| \geq \frac{\epsilon}{(\alpha + 1)m}\right] \leq 2 \exp\left(-\frac{2T(e)\epsilon^2}{(\alpha + 1)^2 m^2}\right).$$

Note that $T(e) \geq \frac{(\alpha+1)^2 m^2}{2\epsilon^2} \log \frac{2m}{\delta}$ gives

$$\exp\left(-\frac{2T(e)\epsilon^2}{(\alpha + 1)^2 m^2}\right) \leq \frac{\delta}{2m}.$$

Therefore, by taking a union bound, we have

$$\Pr\left[|\widehat{s}(e) - s(e)| < \frac{\epsilon}{(\alpha + 1)m}, \ \forall e \in E\right] \geq 1 - \delta.$$

By Lemma 10 for $\epsilon := \frac{\epsilon}{(\alpha+1)m}$, we see that $\mathcal{C}_{\text{out}}$ is an $(\alpha, \epsilon)$-approximate solution for instance $(V, s)$ of the offline problem minimizing (1) w.p. at least $1 - \delta$, as desired. $\square$

The next proposition evaluates the performance of Algorithm 7.

**Proposition 2.** *Given a sampling budget $T$ and additive error $\epsilon \in (0, 0.5)$, the uniform sampling algorithm with an $\alpha$-approximation oracle for the FB setting (Algorithm 7) outputs $\mathcal{C}_{\text{out}}$ that satisfies*

$$\Pr\left[\text{cost}_\text{s}(\mathcal{C}_{\text{out}}) > \alpha \cdot \text{OPT}(\text{s}) + \epsilon\right] = \mathcal{O}\left(n^2 \exp\left(-\frac{T\epsilon^2}{\alpha^2 n^6}\right)\right).$$

*Proof.* As $e \in E$ has been pulled at least $\lfloor \frac{T}{m} \rfloor$ times, by Lemma 1, we have

$$\Pr\left[|\widehat{\text{s}}(e) - \text{s}(e)| \geq \frac{\epsilon}{(\alpha + 1)m}\right] \leq 2\exp\left(-\frac{2T\epsilon^2}{(\alpha + 1)^2 m^3}\right).$$

Taking a union bound for all $e \in E$, we have

$$\Pr\left[|\widehat{\text{s}}(e) - \text{s}(e)| < \frac{\epsilon}{(\alpha + 1)m}, \ \forall e \in E\right] \geq 1 - 2\sum_{e \in E} \exp\left(-\frac{2T\epsilon^2}{(\alpha + 1)^2 m^3}\right)$$

$$\geq 1 - 2m \exp\left(-\frac{2T\epsilon^2}{(\alpha + 1)^2 m^3}\right).$$

By Lemma 10, when for all $e \in E$, $|\widehat{\text{s}}(e) - \text{s}(e)| < \frac{\epsilon}{(\alpha+1)m}$, we have $\text{cost}_\text{s}(\mathcal{C}_{\text{out}}) \leq \alpha \cdot \text{OPT}(\text{s}) + \epsilon$, which concludes the proof.

$\square$

