# OpenReview forum: "Query-Efficient Correlation Clustering with Noisy Oracle"
_NeurIPS.cc/2024/Conference — NeurIPS 2024 poster_

### Official Review · Reviewer_RiHB · 2024-06-26

**Soundness:** 3
**Presentation:** 3
**Contribution:** 4
**Rating:** 7
**Confidence:** 4

**Summary:**

The paper considers a correlation clustering setting with an unknown similarity function but with the ability to query a noisy oracle, i.e., the oracle may yield noisy feedback instead of the true similarity of the queried pair of objects. In this scenario, the goal is to achieve a reasonable trade-off between the number of queries to the oracle and the cost of the clustering. In the considered setting, the authors introduce two novel problem formulations within the paradigm of pure exploration of combinatorial multi-armed bandits (PE-CMAB): fixed confidence and fixed budget settings. For both problems, the paper introduces two novel algorithms (KC-FC and KC-FB) and analyzes their theoretical guarantees. Some experiments on real-world graph datasets with some artificially generated features validate the theoretical findings of the work by demonstrating that in both settings the proposed methods outperform a baseline method in terms of sample complexity and cost of the clustering.

**Strengths:**

S1) Good originality: the paper studies two novel problem formulations for correlation clustering with unknown similarity values and an oracle with noisy feedback.

S2) The settings considered in this work are well-motivated in the Introduction.

S3) The paper is overall well-written and very well-contextualized relative to prior work (>80 references) on similar topics.

S4) The proposed algorithms are theoretically analyzed.

**Weaknesses:**

W1) The authors considered the most basic formulation for correlation clustering, where the costs of placing two objects in the same or different clusters sum to 1, and there is a cost for every pair of objects (Equation 1). The authors neither motivate the choice of this specific setting nor discuss the more general case where such costs are not related and some pairs are missing from the objective function. A discussion on how/if the results presented in this paper can be extended to the more general correlation clustering problem would have been appreciated.

W2) Limited number of baselines used in the experiments. Besides their proposed method, the authors only tested against the baseline KwikCluster, which knows the true similarities, and Uniform-FB (Uniform-FC), which is defined by the authors themselves. The lack of comparison with existing methods for general-purpose PE-CMAB algorithms for the FB and FC settings is notable. The authors justify this choice by stating that the theoretical analysis of such algorithms does not hold in the context of correlation clustering considered in this paper (lines 220-227). However, comparing the proposed approach with existing general PE-CMAB approaches would have provided a more convincing argument about the ineffectiveness of existing approaches in practice for the considered problem. This represents a missed opportunity to demonstrate the necessity of their proposed algorithms with practical evidence.

W3) Limited reproducibility: the source code is not provided with the submission.

W4) Missing proof sketch for Theorem 2 in the main paper. As done for Theorem 1, the authors should include a proof sketch for Theorem 2 as well.

W5) The paper is not clear in the following aspects:

- It is not clear why the authors decided to use different methods to generate the similarity values for the FC and FB settings. In particular, why not use the same strategy adopted for the FC setting for the FB setting as well, to observe, by varying Delta_{min}, how often the algorithm yields the desired approximate solution, i.e., estimate the probability in Equation 3?

- The authors specify how the similarity scores, hidden from the learning algorithm, are generated. However, the main paper does not explain how the noisy feedback is provided to the learner when querying a particular pair of objects. Are the defined similarities the mean of a Bernoulli or R-sub-Gaussian distribution? Are the noisy feedbacks generated by sampling from these distributions? Please specify this in the experimental evaluation section.

- Some parameter values are not justified in the paper: why is epsilon set to sqrt{n}, delta = 0.001, and T = n^{2.2}? The latter is said to be for scalability, but why n^{2.2} and not n^{2.1}?

- The motivation for using only instances with n < 1000 nodes for the experiments in Figures 1 and 2 is unconvincing. The authors could have used better hardware for the experiments, not just a notebook.

- The authors do not run Uniform-FC algorithms in the experiments but report the sample complexity, which can be calculated without running it. How is this done? This is not clear and should be explicitly stated on line 325.

Minors:
- The font size in Figures 1, 2, and 3 is too small and difficult to read.
- Algorithm 2, line 7: please specify that the updates are done for edges \bar{e}_t^g and \bar{e}_t^b to improve clarity.
- State in the paragraph "Algorithm details" that the sets G and B correspond intuitively to the sets of "good" and "bad" pairs, respectively. This is only stated in the algorithm.
- Line 323, “as it will be shown later” is too general. Please be more specific (using pointers) if it refers to something shown in the Figures or the Supplemental Material.

**Questions:**

Q1) In the more general case of correlation clustering (see W1), the KwikCluster algorithm does not provide any theoretical guarantees (even in the offline setting), thus the theoretical guarantees of KC-FC and KC-FB algorithms should not hold either, is it correct? If yes, how can your proposed algorithms be modified to account for such a general setting?

Q2) Besides the missing theoretical guarantees of existing “general-purpose” PE-CMAB on the specific PE-CMAB instance considered in this paper, are there any other reasons that make the applicability of such algorithms unfeasible in practice in the context of the problems considered in this paper? (See also W2)

Q3) How is the noisy feedback provided to the learner in the experiments when querying a particular pair of objects?

Q4) How is the sample complexity of the Uniform-FC algorithm computed without running it?

Q5) In the experiments, why is epsilon set to sqrt(n), delta to 0.001, and T to n^2.2?

Q6) Why did you decide to use different methods to generate the similarity values for the FC and FB settings. As an example, why not use the same strategy adopted for the FC setting for the FB setting as well, to observe, by varying Delta_{min}, how often the algorithm yields the desired approximate solution, i.e., estimate the probability in Equation 3?

**Limitations:**

Yes

---

> ### Author Rebuttal · Authors · 2024-08-05
>
> We sincerely appreciate your review.
>
> **W1 and Q1:**
> Thank you for your insightful question, which is precisely aligned with the critical discussions we had during the process of this work.
>
> As you mentioned, our offline problem (minimizing the cost function (1)) is one of the most basic formulations for Correlation Clustering (CC), which was introduced by the seminal work of Bansal et al. [7], where CC itself was first proposed. The model is versatile, where the similarity is characterized in a value ranging in $[0,1]$. Owing to the importance, the model has extensively been studied in the literature (see e.g., Ailon et al. [3] and Chawla et al. [18]). In our revised version, we will clarify the position of this model. Thank you for your suggestion!
>
> The theoretical guarantees of KC-FC and KC-FB are based on the approximation ratio of 5 of KwikCluster for the above model, and therefore, they do not hold for the general setting you suggested (i.e., our learning model with the general MinDisagree).
>
> Extending our learning model to the above general setting you suggested or even to other NP-hard problems (not necessarily clustering problems) is definitely an interesting direction. Our study shows that, while most existing PE-CMAB algorithms rely on exact algorithms tied to offline optimization problems, we successfully address an NP-hard problem by fully leveraging the properties of the offline algorithm, i.e., KwikCluster's thresholding property, to propose an online learning algorithm that obtains instance-dependent upper bounds.
>
> Alternatively, another potential approach to deal with a learning model with an offline NP-hard problem could be estimating the gap between the objective values of the worst $\alpha$-approximate solution and the best non-$\alpha$-approximate solution to adapt LUCB (Lower Upper Confidence Bound)-type algorithms in the FC setting or successive reject algorithms in the FB setting. However, estimating such a gap seems infeasible, and thus, designing a stopping condition or rejection strategy in such cases is not straightforward. Dealing with a learning model with an offline NP-hard problem is regarded as one of the major open problems in the field of PE-CMAB.
>
> **W2 and Q2:**
> Existing PE-CMAB methods for both the FC and FB settings require exponential runtime, which is infeasible in practice. Moreover, in the FC setting, even the stopping condition of the algorithm is invalid. As highlighted in Lines 103–109 and Lines 282–289, we emphasize that such existing PE-CMAB methods only work for the case where the offline problem can be solved exactly in polynomial time. However, this is not the case for CC. In fact, existing PE-CMAB methods essentially compute two candidate solutions and compare their goodness to select the currently empirically best solution as the final output, but this strategy cannot be used when only approximate solutions are available.
>
> Specifically, in the FC-setting, existing methods [23, 25, 35, 82] use the LUCB type strategy, and its stopping condition requires the exact computation of the empirical best solution and the second empirical best solution to check if the current estimation is enough or not. When we only have an approximate oracle (i.e., approximation algorithm), such existing stopping conditions are no longer valid, and the algorithm is not guaranteed to stop.
>
> In the literature of combinatorial bandits, the FB setting presents even more computational challenges and a scarcity of results. In the FB setting, the existing successive-reject type algorithm [6, 25, 35] cannot handle partition structures in CC, and require exponential running time.
>
> Our algorithms are the first polynomial-time algorithms that work for the case of PE-CMAB where the underlying offline optimization is NP-hard.
>
>
> **W3:**
> We will make our source code publicly available.
>
> **W4:**
> We will include the proof sketch of Theorem 2, which is based on Lemma 9 and Lemma 8, to the main text.
>
> **W5 including Q3–Q6:**
>
> - [To Q6] The dependence of the sample complexity on the minimal reward gap $\Delta_\min^{-2}$ (as proved in Theorem 1) in the FC setting and the dependence of the error probability on $\exp(−T)$ (as proved in Theorem 2) in the FB setting are statistically significant characteristics. Evaluating the algorithm by varying $\Delta_\min$ in the FC setting and the budget $T$ in the FB setting is a standard experimental setup in the PE-MAB literature to provide empirical evidence for theoretical results. Therefore, we follow this practice.
> - [To Q3] While our model is capable of dealing with either type of observations (as noted in Lines 154–156), we used a Bernoulli distribution in our experiments, for simplicity. We will specify this in the experimental section.
> If we allow each element to mistake a constant number of pairs compared to OPT, then $\epsilon = O(n)$. However, this may be too many errors for practical applications. Therefore, we set a stricter threshold, allowing each element to make only $1/\sqrt{n}$ mistakes, which corresponds to a stricter scale of $\epsilon = O(\sqrt{n})$ used in experiments.
> - [To Q5] Taking larger confidence level $\delta \in (0,1)$ means easier setting; we set it to a small enough, i.e., order of $10^{-3}$ following a standard choice in PE-MAB. Regarding the setting of $T$, even on the largest dataset Wiki-Vote, if we employ $T=n^{2.1}$, KC-FB and Uniform-FB query each pair only four times in their first iterations, which allows for randomness too much, and thus we employed $T=n^{2.2}$, where they query it more than ten times.
> The experiments were performed using a desktop, which has moderate computational resources.
> - [To Q4] As written in Lines 312–313, the pseudocode of the baselines and full analysis are given in Appendix E. Given the Line 1 of Algorithm 5, we can see the precise number of samples that Uniform-FC uses.
>
> **Minors:**
> We will incorporate all of them accordingly. Thank you for your careful reading!

---

> > ### Comment · Reviewer_RiHB · 2024-08-08
> > **Follow-up discussion**
> >
> > I appreciate the detailed response from the authors, which clarified most of my concerns, especially the one concerning W2/Q2.
> >
> > Regarding W1, I understand that the theoretical guarantees of KC-FC and KC-FB rely on the approximation ratio of 5 provided by KwikCluster and thus do not extend to the general setting of correlation clustering. I have a follow-up question: would it be problematic in your analysis to work with a non-constant approximation guarantee for the noisy oracle? If not, extending your work to the general setting might be more feasible by building the algorithmic framework around, for example, the algorithm proposed by Demaine et al. [34], which offers $O(\log n)$ approximation guarantees for general correlation clustering by solving a linear programming relaxation. Admittedly, this would necessitate a more complex algorithmic design and analysis, as, in your response to Reviewer S1WA, you noted that designing an online algorithm when the offline algorithm solves a linear programming relaxation of the problem is quite challenging. Specifically, when using LP-based approximation algorithms, it becomes essential to evaluate the optimal value of the LP—estimated under uncertainty—against the true optimal value, which presents a significant analytical challenge. However, I am curious why this represents a significant challenge and how it differs from the challenge that arises in every CMAB problem, where it is necessary to relate the optimal value of an instance with uncertainty (provided by the mean estimates of the arms) to the true optimal value.

---

> > > ### Author Response · Authors · 2024-08-09
> > > **Challenges in Pure Exploration for NP-hard problems**
> > >
> > > We appreciate you taking the time to review our rebuttal thoroughly and adding the discussion with your constructive questions!
> > >
> > > >  I have a follow-up question: would it be problematic in your analysis to work with a non-constant approximation guarantee for the noisy oracle? If not, extending your work to the general setting might be more feasible by building the algorithmic framework around, for example, the algorithm proposed by Demaine et al. [34], which offers 𝑂(log⁡𝑛) approximation guarantees for general correlation clustering by solving a linear programming relaxation. Admittedly, this would necessitate a more complex algorithmic design and analysis, as, in your response to Reviewer S1WA, you noted that designing an online algorithm when the offline algorithm solves a linear programming relaxation of the problem is quite challenging. Specifically, when using LP-based approximation algorithms, it becomes essential to evaluate the optimal value of the LP—estimated under uncertainty—against the true optimal value, which presents a significant analytical challenge.
> > >
> > > The non-constant approximation ratio itself is not problematic in the analysis. Rather, the behavioral property of an algorithm employed is crucial. Even if KwikCluster admitted only $\alpha$-approximation ratio (e.g., $O(\log n)$) for our offline problem, KC-FC/KC-FB would also admit an $\alpha$-approximation. The reason why KC-FC and KC-FB are able to solve the PE-CMAB problems with theoretical guarantees is that they leverage the property that by accurately estimating the mean of the base arms (i.e., pairs of elements), we can maintain the approximation guarantee in the offline setting, as shown in Lemma 5 for the FC setting and Lemma 9 for the FB setting. However, as you pointed out, building upon the LP-based $O(\log n)$-approximation algorithm in the PE-CMAB setting would present significant analytical challenges, because the region-growing algorithm does not have any desired property, unlike KwikCluster.
> > >
> > >
> > > >However, I am curious why this represents a significant challenge and how it differs from the challenge that arises in every CMAB problem, where it is necessary to relate the optimal value of an instance with uncertainty (provided by the mean estimates of the arms) to the true optimal value.
> > >
> > > Regret minimization in CMAB is quite different from Pure Exploration when working with approximation oracles (i.e., offline approximation algorithms) for solving NP-hard problems. For regret minimization, we can incorporate approximation oracles with the UCB framework, consistent with the optimization under uncertainty principle (Chen et al. [26, 27]; Wang and Chen [81]). However, for Pure Exploration, the main difficulty working with approximation oracles lies in determining the stopping condition --- unlike the optimal value,  objective values of $\alpha$-approximate solutions are not unique, causing it difficult to decide whether the Pure Exploration algorithm has already found a sufficiently good solution to terminate. When an exact computation oracle is available for an offline problem, the use of LCB and UCB scores with exact solutions can set a stopping condition, as seen in many existing LUCB-type approaches in the FC setting. However, this approach becomes invalid when dealing with $\alpha$-approximate oracles. In the FB setting, the Combinatorial Successive Accept Reject algorithm proposed by Chen et al. [24] iteratively solves the so-called Constrained Oracle problem, which is often NP-hard, as later addressed in Du et al. [35]. We anticipate a similar NP-hard problem in our Correlation Clustering problem, requiring a different approach.
> > >
> > > We hope that the explanation addresses your additional questions. We intend to incorporate a summary of these discussions into the revised version.

---

> > > > ### Comment · Reviewer_RiHB · 2024-08-11
> > > >
> > > > I sincerely thank the authors for the further discussion and clarifications. The rebuttal has strengthened my confidence in the quality of this work, and as a result, I have raised my score.

---

### Official Review · Reviewer_7MDd · 2024-07-06

**Soundness:** 3
**Presentation:** 3
**Contribution:** 2
**Rating:** 6
**Confidence:** 2

**Summary:**

**[Setting]**
This paper studies correlation clustering using a noisy oracle. The learner queries a pair of items and receives a noisy estimate of their similarity in the range [0, 1] from an oracle. The goal is to either:
1. [Fixed confidence setting] Minimize the number of queries and return a clustering with cost at most $\alpha OPT + \epsilon$ with high probability
2. [Fixed budget setting] Minimize the probability of returning a clustering with cost more than $\alpha OPT + \epsilon$ for a given query budget $T$

**[Contributions]**
1. An algorithm (KC-FC) for the fixed confidence setting that makes roughly $\tilde{O}(n^2/\Delta_1^2)$ queries ($\Delta_1$ being a custom gap-based metric) and returns a clustering with cost at most $5 OPT + \epsilon$ with high probability.
2. An algorithm (KC-FB) for the fixed budget setting that makes $T$ queries and returns clustering with cost at most $5 OPT + \epsilon$ with probability at least $1 - O(n^3 \exp(-2T\Delta_2^2/n^2))$, where $\Delta_2$ is another custom gap-based metric.
3. Empirical validation that the algorithms work as expected

**Strengths:**

1. This paper considers an oracle that returns values in range [0, 1] instead of a binary response, as in existing works.
2. Results for both fixed confidence and fixed budget settings are included.
3. The paper is well-written and easy to follow.
4. Experiments show that the cost of recovered clustering in the fixed-confidence setting is roughly the same as the cost of clustering obtained from KwikCluster if it is given the "ground-truth" similarity scores.

**Weaknesses:**

1. I am not entirely convinced by the strength of the theoretical results in the fixed confidence setting. Why not move sampling inside the clustering loop in Algorithm 1? Keep sampling $\lbrace u, v_r \rbrace$ for all unclustered nodes $u$ and the chosen pivot $v_r$ until $s(u, v_r) > 0.5 - \epsilon$ or $s(u, v_r) < 0.5 + \epsilon$ can be determined with high confidence. Then move nodes into the same cluster as $v_r$ if $s(u, v_r) > 0.5 - \epsilon$. This would require $\tilde{O}(n / \Delta^2)$ queries per round $r$, leading to overall complexity of $\tilde{O}(nk / \Delta^2)$ instead of $\tilde{O}(n^2 / \Delta^2)$. Algorithm 3 already does something similar.
2. The results will benefit from more context. For example, when $T = \theta(n^2)$, the fixed budget bound becomes meaningless. How does this result compare to the case of noisy correlation clustering with only one sample per edge (e.g., Mathieu and Schudy, 2010 and similar results)? In the absence of a lower bound, such context would be very valuable.
3. More context can also be added to experiments. For example, it would be useful to threshold the oracle's response and compare the query complexity with an approach for binary responses? The naïve baselines seem too simplistic.

**Questions:**

Please respond to the points under weaknesses

**Limitations:**

Yes, the authors addressed the lack of a lower bound.

---

> ### Author Rebuttal · Authors · 2024-08-05
>
> We sincerely appreciate your review.
>
> > I am not entirely convinced by the strength of the theoretical results in the fixed confidence setting. Why not move sampling inside the clustering loop in Algorithm 1? Keep sampling $\{u,v_r\}$ for all unclustered nodes $u$ and the chosen pivot $v_r$ until $s(u,v_r)>0.5-\epsilon$ or $s(u,v_r)<0.5+\epsilon$ can be determined with high confidence. Then move nodes into the same cluster as $v_r$ if $s(u,v_r)>0.5-\epsilon$. This would require $\tilde{O}(n/\Delta^2)$ queries per round $r$, leading to overall complexity of $\tilde{O}(nk/\Delta^2)$ instead of $\tilde{O}(n^2/\Delta^2)$. Algorithm 3 already does something similar.
>
> Thank you for your insightful question. Your observation is correct; we have also considered the procedure you suggested during this work. However, we have arrived at the current form to ensure the theoretical guarantees.
> While incorporating TB-HS within the loop, as you suggested, could potentially reduce the overall sample complexity, we must consider the worst-case scenario where the total number of loops (denoted by $k$ above) can be as large as $n$, and the sample complexity would depend on the worst-case instance gap. As for the latter, more precisely, if we employ the algorithm you suggested, the sample complexity for each loop will be characterized by the gap defined by the similarities between the pairs of $v_r$ and its unclustered neighbors​. In this context, the input to TB-HS will become a random variable. Additionally, randomization in the pivot selection is essential to guarantee the approximation ratio of KwikCluster. Overall, in the resulting analysis, which excludes random variables, the bound will depend on the worst-case instance gap, meaning that the above modification does not improve upon Theorem 1.
>
> > The results will benefit from more context. For example, when $T=\theta(n^2)$, the fixed budget bound becomes meaningless. How does this result compare to the case of noisy correlation clustering with only one sample per edge (e.g., Mathieu and Schudy, 2010 and similar results)? In the absence of a lower bound, such context would be very valuable.
>
> Our FB setting of PE-CMAB formulation and the planted noise models in Mathieu and Schudy [64] are not directly comparable:
> - As discussed in Section 1.2, "Correlation clustering with noisy input" (Lines 111–124), the planted noise model proposed by Mathieu and Schudy [64] (2010) assumes the existence of a ground-truth clustering, where similarities are binary, and labels are flipped with a probability $p$. Makarychev et al. [62] (2015) extended this to general weighted graphs, proposing a model where edge labels ($+$ for similar pairs and $-$ for dissimilar pairs) can flip. However, their model assumes that all similarities $\mathrm{s}(e)$ are known, which is a key difference from our model. Additionally, their model assumes the existence of a ground-truth clustering, and does not discuss sequential selection strategies.
> - In contrast, we do not assume the existence of such a ground-truth clustering. Instead, we consider a scenario where the similarities, represented by edge weights, are completely unknown, and where only noisy evaluations are available through sequential queries. Furthermore, the noisy feedback from all edges is probabilistic, and we consider models where this feedback is independently sampled from distributions such as a sub-Gaussian or Bernoulli distribution. Consequently, these models are not directly comparable; the planted noise models mentioned above do not incorporate sequential algorithmic elements, nor do they address the estimation of weights or the uncertainties present in our model. Our work is based on the PE-CMAB formulation within a sequential learning framework, where the total budget $T$ is naturally defined as being greater than the number of base arms (e.g., number of edges in the graph). This is consistent with all PE-MAB formulations in the FB setting.
>
>
> > More context can also be added to experiments. For example, it would be useful to threshold the oracle's response and compare the query complexity with an approach for binary responses? The naïve baselines seem too simplistic.
>
> Performing KwikCluster on the rounded values obtained by taking the sample mean from a Bernoulli distribution and rounding it to 0 or 1 using the threshold of $0.5$ is an approach that has already been employed in our baselines and even our proposed methods for both the FC and FB settings. If you have another algorithm in mind or specific suggestions, please share the details. We would be happy to address them in our next response.

---

> > ### Comment · Reviewer_7MDd · 2024-08-12
> >
> > Thank you for taking time to address my questions.
> >
> > **Regarding Algorithm1:** Am I correct in saying that moving the sampling step inside the loop does not improve the current $O(n^2 / \Delta^2)$ bound **in the worst case** when $k=n$? If so, isn't the $O(nk/\Delta^2)$ bound doing the same thing while being a bit more general? I also did not completely understand your argument about sample complexity depending on worst-case instance gap. Theorem 1 already depends on $\Delta_{\min}$. Not sure why moving sampling inside the loop will change it. Therefore, while I agree that the modification does not change the worst-case sample complexity, it is more general and allows the algorithm to perform better when $k << n$.
> >
> > **Regarding adding more context:** I appreciate your clarification regarding Mathieu and Schudy (2010). While this is not very important, what I meant was adding a statement like "In an easier problem setting where the true similarities follow a specific structure, making $n^2$ observations is enough to bound the error probability by ________. However, the learner pays a price for being in the more general setting, and the error probability is only bounded by ________ after $n^2$ observations."
> >
> > More important perhaps is a practical comparison. Sample edges uniformly at random for $T$ steps and threshold the values to obtain a signed graph. Use [MS10] to cluster this graph and plot clustering cost vs $T$. How does this curve compare to KC-FB in Fig.3? I understand that [MS10] make assumptions that you don't. But this is one way to compare the practical performance of your algorithm with theirs in a more general setting.

---

> > > ### Author Response · Authors · 2024-08-13
> > > **Further Clarification on Sequential Use of TB-HS and Comparison with Mathieu and Schudy [64]**
> > >
> > > Thank you for dedicating your time to review our rebuttal and for your constructive questions.
> > >
> > > > Regarding Algorithm1: Am I correct in saying that moving the sampling step inside the loop does not improve the current $O(n^2/\Delta^2)$ bound in the worst case when $k=n$? If so, isn't the $O(nk/\Delta^2)$ bound doing the same thing while being a bit more general? I also did not completely understand your argument about sample complexity depending on worst-case instance gap. Theorem 1 already depends on $\Delta_\mathrm{min}$. Not sure why moving sampling inside the loop will change it. Therefore, while I agree that the modification does not change the worst-case sample complexity, it is more general and allows the algorithm to perform better when $k$ << $n$.
> > >
> > >
> > > If we utilize TB-HS within the loop while maintaining the $(5,\epsilon)$-approximation guarantee, we obtain the following sample complexity (omitting the log-log factor, for simplicity here) which, as you pointed out, is better than the sample complexity of Theorem 1 when $k << n$:
> > > $$O\left( \sum_{r=1}^{k} \left( \sum_{e \in I_{V_r}(p_r)} \frac{1}{\tilde{\Delta}^2_{e, \epsilon_r^\prime}} \log \left(\frac{n}{\tilde{\Delta}^2_{e, \epsilon_r^\prime} \delta}\right) + \frac{|V_r|}{\max ( \Delta_{\min,r}, \frac{\epsilon_r^\prime}{2})^2 } \right) \right),$$
> > > where $\epsilon_r^\prime:=\epsilon/(12|I_{V_r}(p_r)|)$, $I_{V_r}(p_r) \subseteq E$ represents the set of pairs between the pivot $p_r$ selected in phase $r$ and its neighbors in $V_r$, and $\Delta_{\min,r}:=\min_{e \in I_{V_r}(p_r)} \Delta_e$.
> > > (We recall that $\tilde{\Delta}_{e, \epsilon_r^\prime}$ is defined as in Equation (2).)
> > >
> > > It should be noted that the symbols related to $r$ and the total number of loops $k$, especially instance-dependent gaps $\tilde{\Delta}_{e, \epsilon_r^\prime}$, are all random variables and complicated. Also, It is not common practice to present a sample complexity with random variables remaining.
> > >
> > > In contrast, the current Theorem 1 does not contain any random variables. Specifically, the significant term related to $\log \delta^{-1}$ is characterized by the gap $\tilde{\Delta}_{e,\epsilon}$ or $\Delta_e$, which represents the distance from 0.5 and not a random variable.
> > >
> > > As we appreciate the suggested variant of our algorithm and its analysis, we will include the above discussion as a remark of Theorem 1 in the revised version. Thank you again for your helpful suggestion.
> > >
> > >
> > > > More important perhaps is a practical comparison. Sample edges uniformly at random for $T$ steps and threshold the values to obtain a signed graph. Use [MS10] to cluster this graph and plot clustering cost vs $T$. How does this curve compare to KC-FB in Fig.3? I understand that [MS10] make assumptions that you don't. But this is one way to compare the practical performance of your algorithm with theirs in a more general setting.
> > >
> > > Mathieu and Schudy [64] devised two algorithms: MainCluster algorithm and Large Cluster algorithm. The first algorithm solves an SDP-relaxation (more precisely, a doubly-nonnegative programming relaxation) for Correlation Clustering, which obviously does not scale to instances with hundreds of elements. As for the second algorithm, according to their runtime analysis (see Section 5.6 in their paper), it takes $O(n^{12\ell})$ time, where $\ell$ is a positive integer, which is again impractical.

---

> > > > ### Comment · Reviewer_7MDd · 2024-08-13
> > > >
> > > > Thank you again for your response. Adding a remark as you have suggested would be useful. I also understand that comparing with other baseline methods is challenging. I have increased my score from 4 to 6. All the best for your submission ! :)

---

### Official Review · Reviewer_S1WA · 2024-07-07

**Soundness:** 3
**Presentation:** 3
**Contribution:** 3
**Rating:** 7
**Confidence:** 3

**Summary:**

This paper introduces algorithms for correlation clustering with noisy, expensive similarity functions. The authors present two formulations in the PE-CMAB framework: fixed confidence and fixed budget. Their proposed algorithms, KC-FC and KC-FB, combine sampling with KwikCluster approximation.

**Strengths:**

The paper addresses the realistic and challenging scenario of noisy and costly similarity functions, which differs from the traditional assumption that provides the exact similarities value.

The theoretical analysis looks solid. The problem setting is motivated by practical applications where computing similarities are expensive and noisy. The paper provides a strong practical evaluation.

The paper is clearly written and easy to read.

**Weaknesses:**

The paper could benefit from a more detailed discussion of the limitations of the current to achieve 3-approximation.

**Questions:**

What is the main challenge to extend this approach for 3-approximation algorithms?

---

> ### Author Rebuttal · Authors · 2024-08-05
>
> We sincerely appreciate your review.
>
> > The paper could benefit from a more detailed discussion of the limitations of the current to achieve 3-approximation.
> > What is the main challenge to extend this approach for 3-approximation algorithms?
>
> As stated in Lines 35–39, for our offline problem (i.e., the problem of minimizing the cost function (1)), KwikCluster is a 5-approximation algorithm, and it is not known that KwikCluster has a better approximation ratio for the problem. Therefore, our algorithm admits exactly the same approximation ratio as that achieved by KwikCluster for the offline problem, up to an additive error $\epsilon >0$. It is worth noting that when the true similarities are binary, the value of $5$ of the $(5,\epsilon)$-approximation can easily be replaced by $3$, using the approximation ratio of $3$ of KwikCluster for the binary offline problem.
>
> To achieve the $(3,\epsilon)$-approximation in the general case, it seems essential to design an online algorithm that incorporates an approximation algorithm with an approximation ratio of $3$ (or better). For our offline problem, the only algorithm that meets the above condition is the one by Ailon et al. [3] with an approximation ratio of $2.5$, but it seems quite challenging to design an online algorithm incorporating it, because the offline algorithm solves a linear programming relaxation of the problem. When using LP-based approximation algorithms, it becomes necessary to evaluate the optimal value of the LP with the estimated similarities involving uncertainty, against the true optimal value of the LP, which requires a quite challenging analysis.
>
> In our revised version, we will feature the above as one of the interesting future directions in the concluding section. Thank you for your suggestion!

---

> > ### Comment · Reviewer_S1WA · 2024-08-14
> >
> > Thank you for the detailed response. I will maintain my score.

---

### Official Review · Reviewer_B6XK · 2024-07-19

**Soundness:** 2
**Presentation:** 2
**Contribution:** 2
**Rating:** 5
**Confidence:** 3

**Summary:**

This paper copes with the problem of active (weighted) correlation clustering when oracle queries are corrupted by random noise.

Authors frame two variants of the problem where high probability guarantees are required: the fixed-confidence and the fixed budget settings; algorithms for the these settings are provided in order.

Theoretical results are supported by numerical experiments.

**Strengths:**

The subject considered in this paper is of interest, as also confirmed by the vast literature on this problem: indeed, correlation clustering (CC) is one of the most popular clustering framework.

The proposed algorithms are simple to understand and easy to implement. One aspect that should be clarified is the precise runtime of KC-FC which I recommend to include in the statement of Theorem 1.

The performance guarantees in Th. 1 is nice in being instance dependent.

**Weaknesses:**

Related works: it is not clear to me why authors only refers to [40] when mentioning the SOTA bounds for active CC in the noiseless setting, when also [15] nature essentially the same performance guarantee.

As for Theorem 1, the guarantees does not recover the 3 apx. factor when the true similarities are binary, instead settles for the much worse 5 apx. factor.

For binary queries, it seems to me that a simple strategy that ask roughly log(n) times queries for each edge queried (e.g.. the choice of kwick cluster) and takes the majority vote, would be equivalent to an algorithm running on the true similarity and should feature a 3 apx. factor. Could authors comment on that? A similar question holds for similarities in [0,1], how about replacing the majority voting above with a simple average, and then running a standard algorithm on the top?

In the numerical evaluation it may be good to consider adding the above baselines in problems where the similarities are binary.

My main concerns revolves around the relevance of the contribution provided by this paper, if you can provide convincing arguments, I'm willing to raise my score.

I'm (partially) satisfied with the authors answer, and increase my score accordingly.

**Questions:**

N/A

---

> ### Author Rebuttal · Authors · 2024-08-05
>
> We sincerely appreciate your review.
>
> > One aspect that should be clarified is the precise runtime of KC-FC which I recommend to include in the statement of Theorem 1.
>
> As stated in Lines 224–227, each iteration of the while-loop of TB-HS takes $O(m)$ steps in a naive implementation or amortized $O(\log
>  T)$ steps if we manage the arms using two heaps corresponding to LCB/UCB values. Moreover, the number of iterations of the while-loop is upper bounded by $O(T)$. Therefore, TB-HS runs in $O(Tm)$ time in a naive implementation or $O(T\log T)$ time if we manage the arms using the two heaps. As KwikCluster runs in $O(m)$ time, those two values directly characterize the overall running time of Algorithm 1. In our revised version, we will add the above to Theorem 1.
>
> > Related works: it is not clear to me why authors only refers to [40] when mentioning the SOTA bounds for active CC in the noiseless setting, when also [15] nature essentially the same performance guarantee.
>
> Historically, Bonchi et al. [11] in 2013 (i.e., the preprint version of Garcia-Soriano et al. [40]) first provided the guarantee of $3\cdot \mathrm{OPT}+O(n^3/T)$. Then, in 2019, Bressan et al. [15] presented essentially the same guarantee. For reference, see the table in Section 1.1 in Bressan et al. [15], where the authors stated “see also Bonchi et al.” for the guarantee. In the current manuscript, we cite Garcia-Soriano et al. [40] because it is the official conference version of Bonchi et al. [11]. In our revised version, we can also refer to Bressan et al. [15], though it has already been cited in the main text.
>
>
> > As for Theorem 1, the guarantees does not recover the 3 apx. factor when the true similarities are binary, instead settles for the much worse 5 apx. factor.
>
> When the true similarities are binary, the value of 5 of the $(5,\epsilon)$-approximation can easily be replaced by 3, using the approximation ratio of 3 of KwikCluster for the binary offline problem. However, we would not highlight this fact in Theorem 1, as the case of binary similarities is not of our interest. As emphasized in Lines 49–61, dealing with only the binary similarities is one of the significant limitations of existing query-efficient correlation clustering algorithms, which indeed initiates our study.
>
> > For binary queries, it seems to me that a simple strategy that ask roughly log(n) times queries for each edge queried (e.g.. the choice of kwick cluster) and takes the majority vote, would be equivalent to an algorithm running on the true similarity and should feature a 3 apx. factor. Could authors comment on that? A similar question holds for similarities in [0,1], how about replacing the majority voting above with a simple average, and then running a standard algorithm on the top?
>
> In the above reviewer’s comments, the definition of “binary queries” is not clear. We suppose that the reviewer considers the case where the true similarities are in $[0,1]$ but the noisy feedback of the oracle follows a Bernoulli distribution. In this case, even if the algorithm suggested by the reviewer performs a sufficiently large number of queries (not necessarily $\log n$ times) for obtaining accurate estimations, the approximation guarantee cannot be 3. This is because we are discussing the approximation guarantee in terms of the cost function (1) defined on the true similarities in $[0,1]$ rather than the similarities determined by the majority voting. It is worth noting that KwikCluster for the offline problem of minimizing the cost function (1) is a 5-approximation algorithm, and it is not known that KwikCluster has a better approximation ratio for the problem. The same discussion applies to the second question.
>
> In case where the reviewer means “binary queries" by the oracle that behaves as follows: for a given $e\in E$, if $\mathrm{s}(e)=0$ holds, the oracle returns 0 with probability greater than $0.5$ and returns 1 otherwise, whereas if $\mathrm{s}(e)=1$ holds, the oracle returns 1 with probability greater than $0.5$ and returns 0 otherwise, the value returned by the oracle no longer follows any distribution with a mean equal to the true similarity value $0$ or $1$. Therefore, this setting is out of our model. It is worth noting that the above model is more similar to those reviewed in Lines 111–124. As for the case of non-binary similarities, the strategy mentioned above has already been employed in our algorithm.
>
> > In the numerical evaluation it may be good to consider adding the above baselines in problems where the similarities are binary.
>
> We note that the algorithm design and the statistical metrics of the algorithms are different across the FB setting (error probability) and in the FC setting (sample complexity). Indeed, in the FB-setting, if one queries $\log n$ times for each pair of the pivot and its neighbor at every iteration, it can exceed the given budget $T$, resulting in an invalid algorithm. In the FC-setting, such an algorithm does not have any approximate guarantee on the output while the sample complexity is trivially bounded by $O(n^2 \log n)$. Only algorithms that are $(\epsilon, \delta)$-PAC are of our interest, in line with the standard experimental practices of all other PE-MAB studies.
>
>
> > My main concerns revolves around the relevance of the contribution provided by this paper, if you can provide convincing arguments, I'm willing to raise my score.
>
> We hope that our response has addressed all of your concerns accordingly.

---

### Decision · Program_Chairs · 2024-09-25

**Decision:**

Accept (poster)

**Comment:**

The paper studies a correlation clustering problem where the similarity between each pair of elements can only be obtained by querying a noisy oracle.

The reviewers have mixed opinions on this paper, but generally on the positive side.  Most reviewers agree that the problem is well-motivated, with solid theoretical analysis and practical evaluation.  One reviewer initially had concerns on the strength of the theoretical results in the fixed confidence setting, but the authors' response has successfully changed the reviewer's mind. Two reviewers increased their scores following the authors' rebuttal, which is a positive development.